# EVO-STEP: ITERATIVE PROBLEM GENERATION AND STEPWISE VALIDATION FOR OPTIMIZING LLMS IN OR

## ABSTRACT

Large Language Models (LLMs) have revolutionized various domains but face significant challenges in tackling optimization modeling tasks for Operations Research (OR) problems, particularly when dealing with complex problem. In this work, we propose Evo-Step-Instruct, a framework that augments existing datasets and generates high-quality fine-tuning data tailored to OR modeling tasks. Evo-Step-Instruct employs iterative problem generation to progressively increase problem complexity and stepwise validation to rigorously validate data, preventing error propagation and ensuring the quality of the generated dataset. Leveraging this framework, we fine-tune open-source LLMs, including LLaMA-3-8B and Mistral-7B, to develop Evo-Step—a model that achieves state-of-the-art performance on benchmarks such as NL4OPT, MAMO, and IndustryOR. Extensive experiments demonstrate the superior performance of Evo-Step, especially in addressing complex OR tasks, with a notable 17.01% improvement in micro average accuracy on difficult problems. These findings highlight the effectiveness of combining structured validation with gradual problem refinement to advance the automation of decision-making processes using LLMs. The code and dataset are available at https://anonymous.4open.science/r/Evo-Step-F5AB.

## 1 INTRODUCTION

Operations Research (OR) is a valuable discipline for addressing complex decision-making problems, widely applied in fields such as economics, engineering, and computer science Bertsimas et al. (2019); Pereira et al. (2022); Belgacem et al. (2020). Effective implementation of OR involves two essential steps: modeling real-world problems and solving them. Despite significant advancements in solution techniques and the development of more efficient solvers, the construction of appropriate models remains a considerable challenge. Such a task requires formulating natural language descriptions into precise mathematical models, which is labor-intensive and demands domain-specific expertise as well as a deep understanding of modeling methodologies. These requirements greatly restrict the broader application of OR, particularly in real-world scenarios.

Recent developments in Large Language Models (LLMs) have enhanced the feasibility of automating optimization modeling. Approaches like Chain-of-Experts (CoE) Xiao et al. (2023) and OptiMUS AhmadiTeshnizi et al. (2024) employ well-crafted prompts and multi-agent systems to enhance the construction of optimization models and corresponding programs. However, these approaches rely on general-purpose LLMs, which, though powerful, are not specifically tailored for OR, limiting their effectiveness in addressing specialized challenges. Additionally, the need to upload sensitive data poses additional privacy concerns. In response, ORLM Tang et al. (2024) presents an alternative by fine-tuning open-source LLMs using a dataset of 30K examples generated from 686 industry cases. While this improves the model's performance for OR modeling, ORLM remains semi-automated, requiring significant manual post-processing to achieve satisfactory results. Moreover, its prompt design lacks the precision needed to manage problem complexity and diversity, resulting in suboptimal outputs. Furthermore, modeling errors are not identified in real-time, allowing inaccuracies to persist and propagate. While rule-based post-processing can address minor errors, it often fails to rectify deeper logical and structural issues, further compromising data quality.

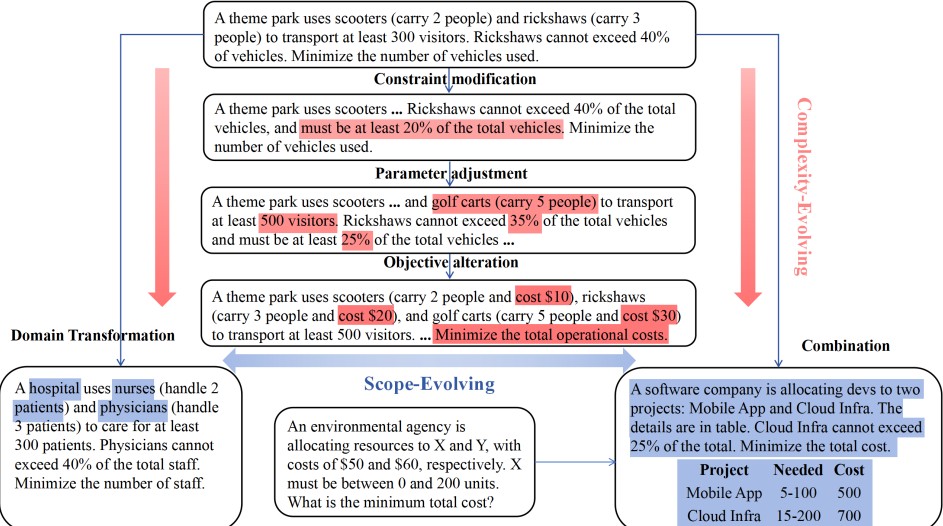

Figure 1: Examples of Iterative Problem Generation. It includes two types of methods: Complexity-Evolving, which refines problem complexity through constraint modification, parameter adjustment, and objective alteration; and Scope-Evolving, which enhances diversity via domain transformations and problem combinations. Red and blue backgrounds indicate changes introduced by Complexity- and Scope-Evolving, respectively. Repeated content is replaced with "..." for clarity.

Utilizing high-quality training data is vital for improving the modeling capabilities of LLMs. However, current methods not only rely heavily on manual post-processing but also struggle to ensure data reliability. To address these limitations, we propose an approach from two primary perspectives. First, we enhance the prompt design and introduce an Iterative Problem Generation, as shown in Figure 1. This method incrementally increases the complexity and scope of the problems, allowing the dataset to retain varying levels of difficulty and breadth. This diversity plays a crucial role in improving the model's generalization capabilities, as WizardLM Xu et al. (2024) suggested. Second, we incorporate a stepwise validation mechanism that performs real-time checks throughout the generation process, effectively filtering out low-quality or erroneous data. This prevents errors from entering and propagating through the seed dataset. We refer to this framework as **Evo-Step**-Instruct. Our framework eliminates the need for post-processing, enabling fully automated generation while reducing API costs by utilizing only high-quality data for future iterations.

Evo-Step-Instruct consists of two key components: Iterative Problem Generation and Stepwise Validation Mechanism. The Iterative Problem Generation is specifically designed to address the unique challenges of OR-specific tasks, such as complex variable definitions and strict constraint implementation. By employing tailored methods such as Complexity-Evolving and Scope-Evolving, this approach generates a dataset enriched with enhanced complexity and diversity from the given one, facilitating the fine-tuning of LLMs to enhance their modeling ability for OR problems. As illustrated in Figure 1, Complexity-Evolving increases the complexity of the problem refining constraints, objectives, or parameters, while Scope-Evolving expands linguistic diversity and problem scope by adapting problems to new contexts or merging scenarios. This approach ensures that the generated dataset captures a wide range of complexities and provides robust coverage.

As new generated problems become increasingly complex, current LLMs often struggle to solve them accurately, resulting in errors. If these errors remain undetected and uncorrected, they will propagate through the iterative process, ultimately affecting the quality of the generated data. To mitigate this, the stepwise validation mechanism is implemented to not only prevent errors but also guarantee the accurate application of essential modeling techniques. Problems are first validated via a description checker for completeness, followed by checks on variables, constraints, and programs. Identified issues are resolved via feedback loops, with advanced techniques like the Big-M method verified using specially designed prompts that guide the LLM step-by-step to confirm accurate implementation. This validation process enables the generation of reliable and high-quality datasets, which are crucial for fine-tuning LLMs and enhancing their modeling ability for OR problems.

In order to evaluate the effectiveness of Evo-Step-Instruct, we collect 260 seed cases and generate nearly 4.5K examples. This data is then applied to train LLaMA-3-8B AI@Meta (2024) and Mistral-7B Jiang et al. (2023), producing a model named Evo-Step. Furthermore, we manually review benchmarks including NL4OPT Ramamonjison et al. (2023), MAMO Huang et al. (2024), and IndustryOR Tang et al. (2024), correcting a large number of examples with error labels. Experiments across these benchmarks indicate that our method outperforms existing approaches, achieving a 6.07% improvement in the micro average and a 7.93% enhancement in the macro average. Notably, when focusing on more complex components, Evo-Step exhibits a more significant advantage, attaining improvements of 17.01% and 12.26% in micro and macro averages, respectively. This substantial lead underscores our method's capability to manage complex problems effectively.

Our contributions are as follows:

• Introduction of advanced feedback mechanisms and real-time data updates, significantly reducing error propagation, thereby eliminating the need for extensive manual post-processing.

• Development of Evo-Step-Instruct, a novel framework specifically designed to enhance the capabilities of open-source LLMs for effectively modeling OR problems.

• Proposal of the Evo-Step model, which achieves state-of-the-art performance across several benchmarks and particularly for complex problems, with additional manual corrections applied to errors in established benchmarks such as NL4OPT, MAMO, and IndustryOR.

## 2 RELATED WORK

**LLM-based Automated Modeling for OR** is an emerging field that uses LLMs to generate mathematical models for OR problems. Existing methods can be categorized into prompt-engineering and fine-tuning. Approaches like Chain-of-Thought Wei et al. (2022) and Reflexion Shinn et al. (2024) improve performance but are not specialized for OR. More advanced methods, including OptiGuide Li et al. (2023a), Chain-of-Experts Xiao et al. (2023), and OptiMUS AhmadiTeshnizi et al. (2024), employ multi-agent systems with GPT to construct models but encounter difficulties with complex problems due to GPT's limitations. ORLM Tang et al. (2024), conversely, utilizes a large dataset generated from industry cases and GPT-4, coupled with rule-based post-processing, to fine-tune LLMs and improve outcomes. However, it lacks precise prompt design and effective filtering mechanisms. Our framework addresses these limitations by iterative-based generation and real-time validation to control complexity and minimize errors, thereby enhancing performance.

**Data Augmentation** improves LLM performance by generating synthetic datasets, often used when real-world data is insufficient for complex tasks Wang et al. (2022); An et al. (2023); Gandhi et al. (2024); Oh et al. (2023); Xu et al. (2024); Pan et al. (2023); Zhou et al. (2024). In operations research, data augmentation approaches like Prasath & Karande (2023); Li et al. (2023b) focus on synthesizing optimization problems from natural language descriptions, but with limited complexity. ORLM Tang et al. (2024) expands industry case datasets through modifications and rephrasings, while ReSocratic Yang et al. (2024) takes a reverse data synthesis approach, generating optimization scenarios from solutions. Among all these works, the closest to ours is Evol-Instruct Xu et al. (2024), which uses In-depth Evolving and In-breadth Evolving to generate instruction data. However, as OR modeling presents unique challenges, we propose a stepwise validation mechanism to ensure accuracy and avoid error propagation in generated data.

## 3 METHOD

This section outlines the proposed framework Evo-Step-Instruct. As depicted in Figure 2, It comprises two primary components: generators and a stepwise validation mechanism. The details of the generators are provided in Sec. 3.2, while the stepwise validation mechanism is detailed in Sec. 3.3.

### 3.1 PRELIMINARY

We start the generation from a given initial dataset, denoted as $D = \{(q_i, m_i)\}_{i=1}^{K}$, where each instance includes a problem description $q_i$ and its associated mathematical model and program $m_i$.

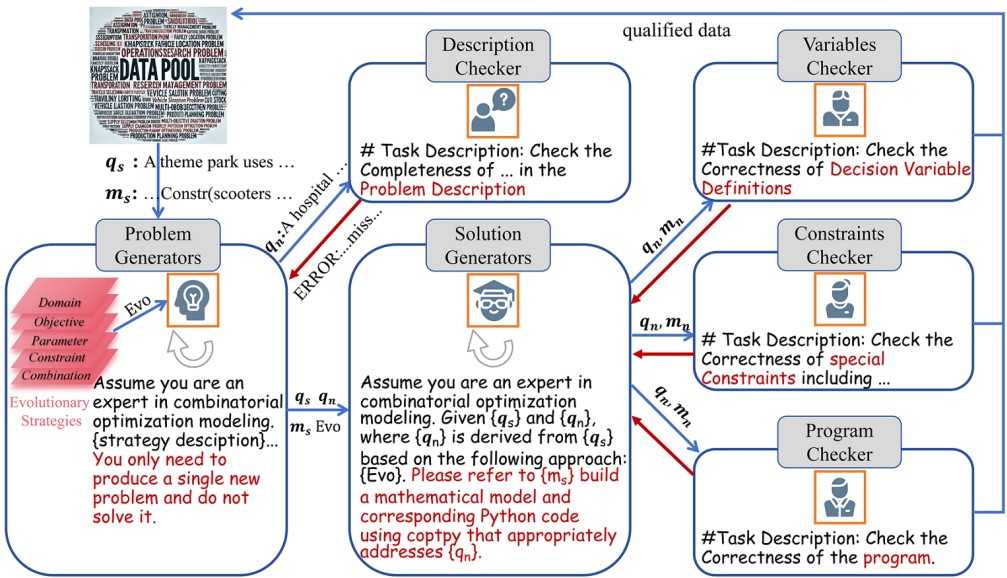

Figure 2: The framework of Evo-Step-Instruct. Each iteration begins by sampling seed data from an initial dataset. The Problem Generator employs evolving prompts to create a problem description, which is reviewed by the Description Checker. Feedback from the checker is used to refine the description until it meets the required standards. Once validated, the Solution Generator produces a solution that undergoes rigorous validation by the Variables, Constraints, and Program Checkers. Any identified errors are returned as feedback for further refinement. Only qualified solutions with descriptions are added to the dataset. All components leverage LLMs with well-designed prompts to ensure high-quality outputs. Red text highlights specific customization requests within the prompts. The collected dataset is used to fine-tune LLMs, enhancing their modeling capabilities.

A qualified $q_i$ must contain an objective function, constraints, and all relevant parameters with specified numerical values. The model $m_i$ implements the constraints and objective functions defined in $q_i$ and generates executable code. An example of the training data is provided in Appendix A.1. The parameter $K$ denotes the size of the initial seed dataset.

## 3.2 GENERATORS

The problem generator adopts an iterative problem generation methodology, progressively producing problems with increasing complexity and diversity. In each iteration, a seed data $(q_s, m_s)$ is randomly sampled. A specific evolving method, denoted as $f_e$, is then applied to create a new problem description $q_n = f_e(q_s)$. This process employs prompt-based LLM methods to refine problem descriptions and systematically expand their scope. These methods can be categorized into two types: **Complexity-Evolving** and **Scope-Evolving**, as detailed below.

**Complexity-Evolving** enhances problem complexity by modifying existing conditions or introducing new elements. In response to the specific characteristics of OR problems, three main approaches are included: constraint modification, objective alteration, and parameter adjustment. These methods incrementally raise the complexity while maintaining the problem's logical integrity.

Constraint modification involves revising existing constraints or adding new ones to enhance the problem, with the core principle being to *"modify constraints based on the given problem while retaining its logical structure."* This ensures that the essential logic of the problem remains intact as complexity increases. Similarly, objective alteration either modifies existing objectives or introduces new ones, and we limit that the modifications cannot merely change to coefficients. Parameter adjustment changes values or adds additional elements. These approaches, while tailored to specific contexts, follow the common principle of preserving the underlying structure. Together, they enhance the difficulty of the problem from various perspectives.

---

**Prompt for constraint modification of Complexity-Evolving**

Assume you are an expert in combinatorial optimization modeling.
Modify constraints or add new constraints based on the given problem while retaining its logical structure. Note that the modifications or additions to the constraints should be limited to a maximum of one. The newly generated problems should align with real-world scenarios.
You only need to produce a single new problem and do not solve it.

**Given example1:**  {Here is Example1}

**Given example2:**  {Here is Example2}

**Given input:**  {Here is the original problem description}

**Answer:**

---

Figure 3: Prompt template for constraint modification. This template specifies tasks to modify or add constraints while maintaining logical consistency. Red text highlights a restriction to process only one constraint at a time, ensuring controlled complexity.

Nevertheless, the generated problems may become so complex that they exceed the processing capabilities of LLMs. To manage this, modifications to constraints or objectives are limited to one at a time, and parameter adjustments introduce at most one new entity per iteration. Figure 3 provides an example of a prompt for constraint modification, where the red text highlights the specific limitations applied to manage the increase in complexity. These restrictions ensure a balanced dataset with problems of varying complexity and excludes excessively challenging examples, enhancing the model's generalization capabilities. Additional prompts available in Appendix A.3.

**Scope-Evolving** broadens topic coverage and diversity by transforming the seed example into a different domain or by combining it with another example to create a novel scenario. Domain transformation transfers the fundamental structure of the original problem to a new application domain, while preserving its logical structure and constraints, thereby increasing linguistic and contextual diversity. To ensure practical relevance, we define a list of domains as references. Alternatively, the combination approach merges two distinct problems to create a new one, with the requirement that the resulting problem belongs to a different domain and contains unique details. This approach introduces more significant changes. To control the increased complexity, the new problem is required to be of a similar length to one of the original problems, maintaining manageable difficulty. The prompt templates for Scope-Evolving are provided in Appendix A.4.

As the Complexity- and Scope-Evolving progress, the complexity, scope, and diversity of the generated data expand, ensuring comprehensive coverage across multiple dimensions. Additionally, all the approaches are implemented using two-shot examples to maintain consistency.

**Solution generator** $g$ produces a corresponding mathematical model and program $m_n$ for a valid problem description $q_n$. It generates $m_n = g(q_n, q_s, m_s, f_e)$ by using $q_s$, $m_s$ and evolving method $f_e$ as references. Since LLMs may struggle with complex models, we specifically embed the instruction *"ensuring the format and structure are as consistent as possible with the provided $q_s$ and $m_s$"* directly into the meta-prompt to enforce consistency.

### 3.3 STEPWISE VALIDATION MECHANISM

While the aforementioned generation methods can produce descriptions and solutions, the complexity of OR problem modeling poses significant challenges for current LLMs, often resulting in issues such as missing parameters, ambiguous objectives, or incorrect application of advanced optimization techniques. Without sufficient supervision and error-correction mechanisms, such issues tend to persist, gradually undermining dataset quality and negatively impacting model performance.

To address these challenges, we design a stepwise validation mechanism that performs checks throughout the generation process, eliminating low-quality or erroneous data to maintain dataset

# Task Description: Comprehensive Constraint Validation for OR Problems.

**Important: The checks must be based on the problem description and common sense. No assumptions or conjectures should be made.**
## Solution Description: To verify the correctness of all constraints in the "## Mathematical Model" for an problem, follow this structured approach:

### Step 1: Extract Constraint Definitions
1. In the "## Mathematical Model", identify constraints under "### Constraints."
2. In the "## Python Code Solution Using coptpy", find where "model.addConstr" is used.

### Step 2: Validate Constraint Alignment with Problem Objectives
...
### Step 3: Special Checks on Big-M Method Applications
1. **Absolute Value Constraints:** For constraints of the form $|x_i - x_j| \geq a \ (a \geq 0)$, verify the use of the Big-M method:
- Introduce a binary decision variable $y$ for each constraint, and a sufficiently large constant $M$.
- Split into two constraints: $x_i - x_j \geq a - M * y$ and $x_j - x_i \geq a - M * (1 - y)$
2. **K-Way Selection Constraints:** At most K Selection (N types), confirm constraints are $\sum_{i=1}^{N} y_i \leq K$ and $x_i \leq M * y_i$, where $y_i$ is a binary variable and $M$ is a sufficiently large constant.
...
### Step 4: Confirm Consistency with Python Code
Ensure that the constraints defined in the mathematical model are accurately translated into the code.

If no errors are found: "There are no errors found."
If errors are identified: **Output "ERROR:"** followed by the issue and advice for correction.

Figure 4: Prompt for constraint checker. This prompt provides structured guidance to locate and extract constraints from the problem description and code. Once extracted, constraints are validated for alignment with the problem objectives, and advanced techniques such as Big-M method applications are specifically checked. If no errors are found, a confirmation message is provided; otherwise, errors are reported with corresponding corrective advice.

integrity. This mechanism comprises four checkers, each concentrating on a specific aspect: completeness of descriptions, definition of variables, implementation of constraints, and quality of program. The description checker evaluates whether the generated $q_n$ contains all essential components. If any element is missing, the checker provides feedback, prompting regeneration until validation is successful or the maximum number of attempts is reached. Only after passing this check does the solution generator proceed to produce the mathematical model and program.

Subsequently, additional checkers will cross-reference $q_n$ and $m_n$ to conduct assessments. For decision variables, detailed and step-by-step instructions are offered, along with numerous examples covering common variable types, enabling the checker to ensure the accurate definition of variables.

The constraint checker is responsible for confirming that constraints are formulated correctly and aligned with the problem description. As illustrated in Figure 4, the checker follows a systematic process, first identifying the constraints and then verifying their consistency with the problem's requirements, much like the variable validation process. While all constraints are rigorously reviewed, particular attention is given to advanced techniques such as the Big-M method for absolute value and K-way selection constraints. These examples serve as illustrations of specialized checks, with other advanced techniques also applicable. Afterward, the program checker extracts and executes the program, capturing outputs or errors, and providing feedback to the solution generator as needed.

When errors are identified in $m_n$, they are relayed back to the solution generator, accompanied by the prompt: *"Please regenerate the solution based on the 'Error'. Ensure that the new solution correctly addresses the problem while maintaining the format and structure, with only the necessary corrections and improvements."* The revised solution is then subjected to further testing until

it passes all validation stages. If the maximum number of retries is reached, the problem will be discarded. This comprehensive validation process ensures that both $q_n$ and $m_n$ are free from errors. Only data that successfully pass all assessments will be integrated into the dataset $D$ for future iterations. This approach minimizes errors within $D$, thereby preventing the propagation of inaccuracies in future generations and safeguarding the overall quality of the collected dataset. Details of the checkers and regeneration process can be found in Appendix A.5

Table 1: Performance Comparison of Various Methods

| Method | NL4OPT | MAMO EasyLP | MAMO ComplexLP | IndustryOR | Micro Avg | Macro Avg |
|---|---|---|---|---|---|---|
| *GPT-3.5* | | | | | | |
| Standard | 13.06% | 35.58% | 10.90% | 6.49% | 24.64% | 16.51% |
| CoT | 33.06% | 66.56% | 13.27% | 12.99% | 46.67% | 31.47% |
| Reflexion | 43.67% | 67.64% | 14.22% | 15.58% | 49.79% | 35.28% |
| CoE | 52.24% | 61.81% | 17.06% | 18.18% | 49.03% | 37.32% |
| *GPT-4* | | | | | | |
| Standard | 72.65% | 81.13% | 24.64% | 25.97% | 65.74% | 51.10% |
| CoT | 76.73% | 84.97% | 29.86% | 25.97% | 69.62% | 54.38% |
| Reflexion | 78.78% | 85.12% | 36.02% | 27.27% | 71.05% | 56.49% |
| CoE | 76.73% | 84.36% | 40.28% | 31.17% | 71.48% | 58.14% |
| *Fine-tune* | | | | | | |
| ORLM | 78.37% | 84.20% | 38.39% | 35.06% | 71.65% | 59.01% |
| Evo-Step-Mistral-7B | 72.65% | 82.06% | 52.61% | **40.26%** | 72.15% | 61.90% |
| Evo-Step-LLaMA-3-8B | **84.49%** | **85.28%** | **61.61%** | 36.36% | **77.72%** | **66.94%** |

# 4 EXPERIMENT

## 4.1 DATASET

We assess our method using a range of datasets, encompassing both simple datasets, such as NL4OPT Ramamonjison et al. (2023) and MAMO EasyLP Huang et al. (2024), and more complex ones, including MAMO ComplexLP Huang et al. (2024) and IndustryOR Tang et al. (2024). The answers have been manually revised where necessary, with all modifications thoroughly documented. A set of examples is included in Appendix A.2.

**NL4OPT** originates from the NL4Opt competition at NeurIPS 2022 and comprises 1,101 simple linear programming problems, of which 289 are used for evaluation. We review the solutions and correct 16 instances that contain inaccuracies.

**MAMO** contains two sub-datasets: EasyLP and ComplexLP. Where the easier one contains 652 simple linear programming problems and the other one includes 211 complex problems, all problems are paired with their optimal solutions. We also reviewed these solutions, rectifying 78 inaccuracies.

**IndustryOR** consists of 100 complex OR problems. Notably, many problems in IndustryOR are found to lack essential information or accurate numerical values, leading to the correction of 50 inaccuracies and the removal of 23 instances that do not meet the necessary modeling criteria.

## 4.2 BASELINES

To facilitate a thorough evaluation, we compare our method against several baselines.

**Standard prompt** directly prompt ChatGPT or GPT-4 Achiam et al. (2023) to generate solution.

**CoT (Chain-of-Thought)** Wei et al. (2022) is a prompting technique that encourages the model to generate intermediate reasoning steps leading to the final solution. This method enhances the model's ability to articulate its thought process, potentially resulting in more accurate outputs.

**Reflexion** Shinn et al. (2024) is a strategy that involves multiple attempts to produce a solution, where each attempt incorporates feedback regarding previous errors. The outputs generated are refined based on the output of the program, promoting improved accuracy over successive iterations.

**Chain-of-Experts (CoE)** Xiao et al. (2023) is a multi-agent prompting framework that utilizes collaborative interactions among various LLMs, referred to as "experts" in this context. This collaborative model enhances problem-solving capabilities by incorporating the strengths of different models.

**ORLM** Tang et al. (2024) is a fine-tuned model for which we employ the checkpoint available on Hugging Face [1]. In addition to this, the release includes 3K training examples[2], allowing us to utilize this dataset in our ablation experiments to further fine-tune a LLaMA-3-8B model as a baseline.

To ensure fairness, all methods were evaluated with a temperature parameter of 0 to standardize output variability. For fine-tuned models, greedy decoding was employed in a zero-shot context, selecting the top-1 completion as the solution. Fine-tuning methods, including Evo-Step, exclusively used the COPT solver to ensure alignment with the raw data format. Prompt engineering methods were evaluated using GPT-3.5 (gpt-3.5-turbo-1106) and GPT-4 (gpt-4-turbo-2024-04-09) on both Gurobi and COPT solvers, with the best results reported for each configuration.

### 4.3 DETAILS

To construct the dataset, we begin with 260 examples and perform 8,400 iterations using GPT-4-turbo-0409, resulting in 4,464 examples for fine-tuning. We fine-tune LLaMA-3-8B AI@Meta (2024) and Mistral-7B Jiang et al. (2023) utilizing the LLaMA-Factory framework Zheng et al. (2024) with the Alpaca format template Taori et al. (2023), applying the LoRA technique Hu et al. (2021) for efficient parameter adaptation. In this setup, the input consists of a fixed prompt with a problem description, and the output includes mathematical models and the corresponding programs. Details of the hyperparameters are provided in Appendix A.6. During inference, we employ greedy search in a zero-shot context, setting the maximum generation length to 2,048 tokens.

### 4.4 METRIC

Considering the potential for minor discrepancies in numerical solutions, we define a comparison rule to account for small inaccuracies. Let $o$ represent the output of generated programs from different methods, and $g$ denote the ground truth. The comparison is governed by the following criterion:

$$\left| \frac{o - g}{g + \epsilon} \right| \leq 10^{-4}, \tag{1}$$

where $\epsilon$ is a sufficiently small number to avoid division errors. When $o$ and $g$ satisfy Eq. 1, they are considered equal.

### 4.5 COMPARISON ANALYSIS

As shown in the Table 1, Evo-Steps based on LLaMA-3-8B and Mistral-7B significantly outperform baselines by a large margin. Especially the best-performing Evo-Step, trained on LLaMA-3-8B, achieves state-of-the-art results on all benchmarks. This demonstrates its superior modeling capability. Notably, fine-tuned LLMs exceed the prompt engineering methods on average. However, the differences are less pronounced in the easier datasets, NL4OPT and MAMO EasyLP. The reason lies in the straightforward modeling requirements of these problems, which primarily require the ability to understand problem descriptions—a strength of models like ChatGPT and GPT-4. In contrast, for datasets containing more complex problems, the performance of fine-tuned models significantly improves, greatly exceeding that of prompt engineering methods. This indicates that fine-tuned models possess enhanced modeling capabilities. A prominent example is MAMO ComplexLP, where the performance advantage of Evo-Step-LLaMA-3-8B reaches 21.33%.

To emphasize the distinctions, we further analyze the results across both simple and complex datasets. For simplicity, we select the prompt engineering method based on GPT-4 as the baseline and the best-performing model from Evo-Step. As shown in Figure 5, nearly all methods perform well on simple datasets, with most achieving over 80% accuracy, except for the Standard method. The differences between methods on simple datasets are relatively minor. In contrast, the results

---

[1]https://huggingface.co/CardinalOperations/ORLM-LLaMA-3-8B/tree/main
[2]https://huggingface.co/datasets/CardinalOperations/
OR-Instruct-Data-3K/viewer

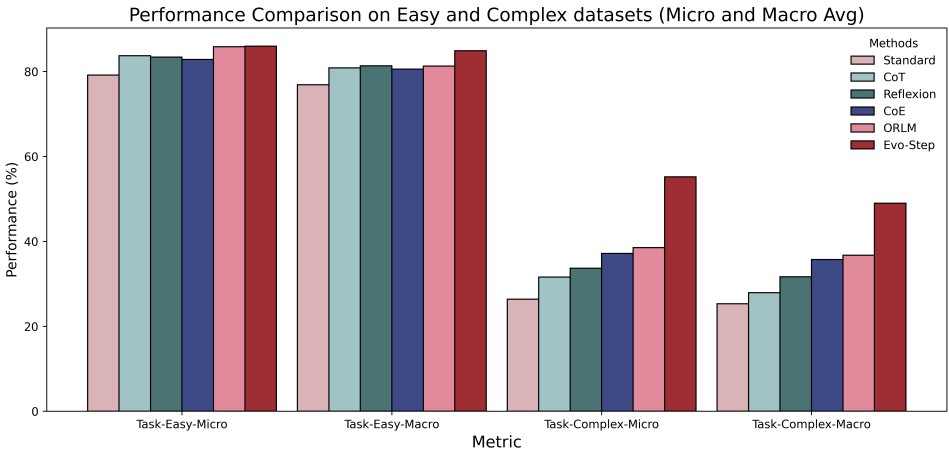

Figure 5: Performance comparison of various methods on easy and complex datasets.

for complex datasets demonstrate that advanced prompt engineering techniques, such as Chain-of-Experts (CoE), significantly outperform Standard, CoT, and Reflexion, though they still lag behind our proposed methods. Notably, Evo-Step achieves an accuracy above 50%, significantly surpassing existing methods and showcasing its superior modeling capabilities for complex problems. Given the intricate nature of complex problem descriptions and the advanced techniques required, our models exhibit a greater capacity to handle higher-order techniques.

## 4.6 ABLATION STUDY

We conduct an ablation analysis to explore the effectiveness of different evolving methods and the composition of the training data, while also facilitating a fair comparison between OR-Instruct and Evo-Step Instruct. For all ablation experiments, we set the hyper-parameter to the same and use LLaMA-3-8B as the backbone. The parameter settings can be found in the Appendix A.6.

Table 2: Ablation Study on different evolving methods

| Method | NL4OPT | MAMO EasyLP | MAMO ComplexLP | IndustryOR |
|---|---|---|---|---|
| Evo-Step | 77.55% | 85.43% | **36.02**% | **23.38**% |
| w/o Constraint Modification | 75.92% | 85.58% | 19.91% | 15.58% |
| w/o Objective Alteration | 77.55% | **85.89**% | 25.12% | 19.48% |
| w/o Parameter Adjustment | 73.06% | 83.59% | 26.07% | 22.08% |
| w/o Domain Transformation | 73.88% | 83.13% | 20.38% | 18.18% |
| w/o Combination | **77.96**% | 85.12% | 33.65% | 22.08% |

**Study on different evolving methods:** Initially, we evaluate the survival rates of examples generated by various methods, yielding the following results: 1,716 for constraint modification, 1,242 for objective alteration, 2,123 for parameter adjustment, 2,077 for domain transformation, and 455 for combination. The higher survival rates for parameter adjustment and domain transformation can be attributed to their relative simplicity, making it easier for examples to pass evaluations. Conversely, the combination is the most challenging, as it requires inputting two sets of descriptions and solutions into the LLM, significantly increasing the likelihood of failure due to potential misalignment. The other two methods, which introduce new elements, are also more prone to errors.

Then, we randomly sample 2,000 examples from datasets without specific methods and train LLaMA-3-8B on this data. The results, presented in Table 2, indicate that excluding domain transformation leads to the poorest performance, with a notable decline observed across all datasets, underscoring its critical importance. While parameter adjustment significantly impacts performance on simpler benchmarks, its effect on complex datasets is less pronounced. In contrast, both constraint modification and objective alteration exert a greater influence on complex datasets compared

to easier ones. Particularly for constraint modification, it introduces additional constraints and increases the difficulty, facilitating the model's ability to process more complex conditions.

Table 3: Comparison of Evo-Step and Evo-Step without mathematical model

| Method | NL4OPT | MAMO EasyLP | MAMO ComplexLP | IndustryOR |
|---|---|---|---|---|
| Evo-Step | **84.49**% | **85.28**% | **61.61**% | **36.36**% |
| Evo-Step-4.73M | 81.22% | 84.97% | 50.24% | 33.77% |
| w/o mathematical model-4.73M | 80.00% | 81.44% | 45.97% | 29.87% |

**Study on the components of training examples :** As described in Sec. 2, each training example includes a mathematical model and corresponding programs utilizing the COPT solver, though only the program is used for problem-solving. To assess the impact of the mathematical model, we remove this component from the entire dataset and train LLaMA-3-8B. The results, presented in Table 3, reveal a significant performance drop upon the removal of the mathematical model. To further mitigate the influence of token count (as data without the mathematical model contain fewer tokens), we maintain a total of 4.73 million tokens across all datasets. Even with equivalent training sizes, the dataset including the mathematical model consistently outperforms the one without it. This improvement can be ascribed to the mathematical model functioning similarly to the Chain-of-Thought approach, providing a structured framework that guides the reasoning process in a systematic manner, effectively bridging the problem description and the code solution. In its absence, the model skips critical reasoning steps, leading to a significant reduction in performance.

Table 4: Comparison of Evo-Step and ORLM with 3K examples.

| Method | NL4OPT | MAMO EasyLP | MAMO ComplexLP | IndustryOR | Micro Avg | Macro Avg |
|---|---|---|---|---|---|---|
| Evo-Step | **78.37%** | 84.51% | **44.08%** | **32.47%** | **72.66%** | **59.86%** |
| ORLM | 75.92% | **88.19%** | 28.91% | 25.97% | 71.05% | 54.75% |

**Comparison of OR-Instruct and Evo-Step Instruct :**ORLM collects 686 industry cases and creates 30,000 examples using the OR-Instruct framework. Among these, 3,000 training examples are made publicly available on Hugging Face. To assess the performance of OR-Instruct in comparison to Evo-Step Instruct, we randomly select 3,000 examples for evaluation. Both datasets, each comprising 3,000 examples, are employed to train LLaMA-3-8B. As illustrated in Table 4, except for MAMO EasyLP, our method uniformly outperforms ORLM, achieving a 1.61% improvement in micro average and a 5.11% enhancement in macro average. The gains on more complex datasets, such as MAMO ComplexLP and IndustryOR, are even more pronounced. These advancements suggest that Evo-Step Instruct possesses superior capabilities and generates higher-quality data, allowing LLMs to more effectively address OR problems, particularly those of greater complexity.

## 5 CONCLUSION

In this paper, we present Evo-Step-Instruct, a novel framework that integrates iterative problem generation with a stepwise validation mechanism to enhance the capabilities of LLMs in addressing complex OR problems. By progressively increasing problem complexity and ensuring data quality through real-time validation, Evo-Step-Instruct effectively prevents error propagation by identifying and removing low-quality data during the generation process. This proactive approach enables full automation without relying on post-processing, ensuring high-quality datasets for fine-tuning. The resulting model, Evo-Step, achieved significant performance improvements across benchmarks such as NL4OPT, MAMO, and IndustryOR, particularly excelling in complex optimization tasks. These results highlight the effectiveness of combining systematic problem generation with structured validation to significantly enhance the modeling capabilities of LLMs.

**Limitations:** The proposed method faces difficulties in dealing with the wide variety of modeling techniques commonly used in OR, which limits its ability to handle the full range of possible scenarios. Moreover, the performance of the approach has not been fully tested across all types of OR problems. Finally, its broader application still needs to be tested in other fields to validate its applicability and adaptability.

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

# A  APPENDIX

## A.1  EXAMPLE FOR TRAINING DATA

---

**Problem Description** $q_i$

A logistics company wants to allocate up to 250 vehicles between operations A and B. To ensure efficiency, the difference between twice the vehicles in A and those in B must be at least 60 units. Additionally, The number of vehicles in A plus twice the number in B should not exceed 300 units. Each vehicle costs $400 for A and $300 for B. What would be the minimum total cost (in dollars)?

- - - - - - - - - - - - - - - - - - - - - - - - - - - - - - - - - - - - -

**Mathematical Model and Program** $m_i$

## Mathematical Model
### Decision Variables:
$x_A, x_B$: Non-negative integer variables representing the number of vehicles allocated to A and B, respectively.
$z$: A binary variable to model the efficiency constraint.
### Objective Function: Maximize: $Z = 400\,x_A + 300\,x_B$
### Constraints:
1. Total number of vehicles constraint: $x_A + x_B \leq 250$.
2. Efficiency condition constraint (Introducing a sufficiently large constant M):
$$2x_A - x_B \geq 60 - M*z, \qquad x_B - 2x_A \geq 60 - M*(1-z),$$
3. Capacity condition: $x_A + 2x_B \leq 300$.
## Program

```python
import coptpy as cp
from coptpy import COPT
env = cp.Envr()
model = env.createModel("LogisticsVehicleAllocation")
x_A = model.addVar(vtype=COPT.INTEGER, name="x_A")
...
model.setObjective(400 * x_A + 300 * x_B, sense=COPT.MINIMIZE)
model.addConstr(x_A + x_B <= 250, name="TotalVehicles")
...
model.solve()
...
```

---

Figure 6: Examples of training data.

We use COPT Ge et al. (2022) as the default solver in our experiments.

## A.2  EXAMPLES FOR MODIFICATIONS OF TEST SETS

**NL4OPT, Entry #228** : Wrong variable definition

Problem: A macro-counting fitness guru only eats salmon and eggs. Each bowl of salmon contains 300 calories, 15 grams of protein, and 80 mg of sodium. Each bowl of eggs contains 200 calories, 8 grams of protein, and 20 mg of sodium. Since the fitness guru has a limit to how many eggs he would like to eat, at most 40% of his meals can be eggs. The fitness guru needs to eat at least 2000 calories and 90 grams of protein. How many of each type of meal should he eat to minimize his sodium intake?

Answer: 430.7692307692307
The answer is initially derived by treating the number of salmon and egg bowls as continuous variables. However, since the number of bowls should be integers, the correct solution is adjusted, and the actual answer is 460.

**MAMO EasyLP, Entry #216** : Incorrect Handling of Absolute Value Constraint

Problem: A retail manager is planning to allocate resources across three different departments: purchasing (X), sales (Y), and logistics (Z). These departments have different cost per unit of resource allocated, with $5 for X, $3 for Y, and $4 for Z. The objective is to minimize the total cost while meeting certain operational constraints. The combined resources allocated to purchasing and sales cannot exceed 1000 units due to budget limitations. Similarly, the combined resources allocated to sales and logistics cannot exceed 800 units due to manpower availability. To ensure a balanced operation, the difference in resource allocation between purchasing and logistics should be at least 200 units. Given that each department has specific bounds on resource allocation (Purchasing can have up to 500 units, Sales up to 300 units, Logistics up to 200 units) and that allocations must be whole numbers due to indivisible nature of the resources being allocated:What is the minimum total cost required for this scenario? type of meal should he eat to minimize his sodium intake?

Answer: 1000
The initial solution was derived without successfully establishing an absolute value constraint for "the difference in resource allocation between purchasing and logistics should be at least 200 units." Instead, only the constraint for one side (greater than or equal to 200) is retained, leading to an error. That is "model.addConstr(x - z ¿= 200, name=ResourceDifferenceConstraint)" in the program. The correct solution, considering both sides of the absolute value constraint, yields an actual minimum total cost of 800.

**MAMO ComplexLP, Entry #216** : Incorrect Handling of Subtour Elimination

Problem: Imagine a logistics manager tasked with planning a delivery route for a truck that needs to visit four different cities to distribute goods. The cities are identified numerically as 1, 2, 3, and 4. The truck can start its journey from any of these cities but must travel to each city exactly once and then return to the starting point. The objective is to arrange this route in such a way that the total travel cost is minimized. The costs associated with traveling between the cities are as follows: The cost to travel from City 1 to City 2 is 52 units, to City 3 is 89 units, and to City 4 is 11 units. From City 2, it costs 52 units to reach City 1, 14 units to get to City 3, and 13 units to City 4. Traveling from City 3, the costs are 89 units to City 1, 14 units to City 2, and 87 units to City 4. Lastly, from City 4, it costs 11 units to go to City 1, 13 units to City 2, and 87 units to City 3. What is the minimum total travel cost for the truck to visit each city exactly once and return to the starting city?

Answer: 50
The initial solution was derived without successfully establishing the subtour elimination constraint for the Traveling Salesman Problem (TSP). As a result, subtours were not eliminated properly, leading to an incorrect minimum total travel cost of 50 units. The correct solution, ensuring that subtours are eliminated and all cities are visited exactly once, yields an actual minimum total travel cost of 127 units.

**IndustryOR, Entry #86**: Missing Numerical Data

Problem: Fighter jets are important combat tools, but in order for them to be effective, there must be enough pilots. Therefore, in addition to a portion of the produced fighter jets being used directly for combat, another portion needs to be allocated for pilot training.

It is known that the number of fighter jets produced each year is $a_j (j = 1, \cdots, n)$, and each fighter jet can train k pilots per year. How should the production of fighter jets be allocated each year to maximize their contribution to national defense over a period of n year?

There is no numerical value for all parameters.

## A.3 Prompt Templates for Complexity-Evolving

### A.3.1 Prompt Templates for objective alteration

---

**Prompt for objective alteration of Complexity-Evolving**

Assume you are an expert in combinatorial optimization modeling. Modify the objective function to either transform the current objective into a different metric or add a new objective to convert it into a multi-objective optimization problem, while retaining its logical structure. The modifications or additions to the objective function should be substantial and not merely changes to coefficients. If there are already two or more objective functions, no new objectives may be added; only the existing objectives can be modified. The newly generated problems should align with real-world scenarios. You only need to produce a single new problem and do not solve it.

**Given example1:**               {Here is Example1}

**Given example2:**               {Here is Example2}

**Given input:**          {Here is the original problem description}

**Answer:**

---

### A.3.2 Prompt Templates for parameter adjustment

---

**Prompt for parameter adjustment of Complexity-Evolving**

Assume you are an expert in combinatorial optimization modeling. Adjust the parameters of the given problem while retaining its logical structure, constraints, and objective. When introducing a new entity, restrict the introduction to at most one new entity to control the complexity of the problem. The newly generated problems should align with real-world scenarios.
You only need to produce a single new problem and do not solve it.

**Given example1:**               {Here is Example1}

**Given example2:**               {Here is Example2}

**Given input:**          {Here is the original problem description}

**Answer:**

---

## A.4 Prompt Templates for Scope-Evolving

### A.4.1 Prompt Templates for Domain transformation

---

**Prompt for domain transformation of Scope-Evolving**

Assume you are an expert in combinatorial optimization modeling.
Transform the basic structure of the given problem into a different application domain while retaining its logical structure and constraints. The new application domain can include, but is not limited to, the following: the following: Education, Manufacturing, Logistics, Retail, Agriculture, IT Services, Healthcare, Event Planning, Construction, Entertainment, Research and Development, Hospitality, Defense, Energy Sector, Transportation, and Telecommunications.
You only need to produce a single new problem and do not solve it.

**Given example1:**               {Here is Example1}

**Given example2:**               {Here is Example2}

**Given input:**          {Here is the original problem description}

**Answer:**

---

## A.4.2 PROMPT TEMPLATES FOR COMBINATION

---

**Prompt for combination of Scope-Evolving**

Assume you are an expert in combinatorial optimization modeling. Given two problems (#Problem1 and #Problem2), generate a new problem. The new problem should be similar in length to one of the original problems but should belong to a different domain and have distinct specific details. The newly generated problem should align with real-world scenarios.
You only need to produce a single new problem and do not solve it.

**Given example1:**                        {Here is Example1}

**Given example2:**                        {Here is Example2}

**Given input:**

**# Problem1:**                        {Here is the first problem description}

**# Problem2:**                        {Here is the second problem description}

**Answer:**

---

## A.5 PROMPT TEMPLATES FOR CHECKERS AND REGENERATION

## A.5.1 PROMPT TEMPLATES FOR DESCRIPTION CHECKER

---

**Check the Completeness of Objective Function and Parameters in the Problem Description**

**Important: The checks must be based on the problem description and common sense. No assumptions or conjectures should be made. The conclusions must be justified by the problem description or common sense.**
## Solution Description:To ensure the problem description contains all necessary information for the objective and that all parameters are specified with values, follow this structured approach:

### Step 1: Extract Constraint Definitions
1. In the "# Question:" section, locate the description of the objective function.
2. Confirm that the objective function is clearly defined, specifying what needs to be minimized or maximized.

### Step 2: Extract Parameter Information
1. Identify all parameters mentioned in the problem description.
2. Ensure that all parameters have specified numerical values or clear definitions in the "# Question:" section.

If there are no errors, output: **"There are no errors found."**
If there are errors, output the specific errors with the format: **Output "ERROR: [description of error]"**. Specify whether it is due to a missing objective or a missing parameter value.

Please check for any missing objective function information and undefined parameter values of **Input** based on the steps above. **Do not repeat the prompt, only provide the errors and their descriptions if any, or confirm there are no errors.**

---

## A.5.2 PROMPT TEMPLATES FOR DECISION VARIABLE CHECKER

---

### Check the Correctness of Decision Variable Definitions

**Important: The checks must be based on the problem description and common sense. No assumptions or conjectures should be made. The conclusions must be justified by the problem description or common sense.**
## Solution Description:To check the definitions of decision variables in the "## Mathematical Model:" for a combinatorial optimization problem, follow this structured approach:

### Step 1: Extract Decision Variable Definitions
1. In the "## Mathematical Model:" section, find definitions under "### Decision Variables."
2. In the "## Python Code Solution Using coptpy:", identify definitions where model.addVar is used.

### Step 2:Confirm Consistency with Problem Description
1. Ensure each variable's type and bounds align with the problem's actual meaning.

### Step 3:Confirm Variable Types and Bounds
**Note: The examples provided below are not exhaustive. Specific examples should be analyzed based on their actual meaning in the context of the problem.**
**Integer Variables (Bounds > 0):** {Examples for Integer Variables}
**Binary Variables (0 or 1):** {Examples for Binary Variables}
**Continuous Variables (0 or 1):** {Examples for Continuous Variables}
**Continuous Variables with Range:** {Examples for Continuous Variables with Range}

### Step 4: Check the Python Code Solution Using coptpy
1. For integer variables: Ensure vtype=COPT.INTEGER.
2. For continuous variables: Ensure vtype=COPT.CONTINUOUS.
3. For binary variables: Ensure vtype=COPT.BINARY.

If there are no errors, output: **"There are no errors found."**
If there are errors, output the specific errors with the format: **"ERROR: [description of error]"** and suggest how to fix them.
Please check for any errors in the variable definitions based on the steps above. **Do not repeat the prompt, only provide the errors and fixes if any, or confirm there are no errors.**

---

### A.5.3 PROMPT TEMPLATES FOR REGENERATING THE PROBLEM DESCRIPTION

---

**Prompt for regenerating the problem description**

The #Problem is a generated problem but has some 'Error'. Please regenerate the problem description based on the 'Error'. Ensure that the new problem follows the same format and structure as #Problem, with only the necessary corrections and detail enhancements. **No solution or any other additional explanations are required.**

#Problem:
{generated_problem}
'Error':
{Error}

**Example1:**

#Problem:
{example1_problem}
'Error':
{example1_error}
'Regenerate':
{example1_regenerate}

**Example2:**

#Problem:
{example2_problem}
'Error':
{example2_error}
'Regenerate':
{example2_regenerate}

**Answer:**

---

### A.5.4 PROMPT TEMPLATES FOR REGENERATING THE SOLUTION



**Prompt for regenerating the solution**

#Solution is the mathematic model and program of #Problem. An 'Error' was detected in #Solution. Please regenerate the solution based on the 'Error'. Ensure that the new solution correctly addresses the problem while maintaining the same format and structure as the original #Solution, **with only the necessary corrections and improvements. No additional explanations are required.**

#Problem:
        {Here is the problem description}

#Solution:
        {Here is the mathematic model and program for #Problem}

'Error':
        {Here is the error}

**Given Example1:**
#Problem:
        {Here is the problem description of Example1}
#Solution:
        {Here is the solution of Example1}
'Error':
        {Here is the error of Example1}

**Given Example2:**
#Problem:
        {Here is the problem description of Example2}
#Solution:
        {Here is the solution of Example2}
'Error':
        {Here is the error of Example2}

**Answer:**



### A.6 HYPER-PARAMETERS FOR TRAINING EVO-STEP AND BASELINES

All experiments are conducted on a single GPU server equipped with eight A100 GPUs, each with 40GB of memory. In experiment, we report the best results of all checkpoints. The maximum token is limited to 2,500. The hyper-parameters for training Evo-Steps are as follows:

Table 5: Hyper-parameters for Training Evo-Steps.

| Backbone | BatchSize Per GPU | Gradient Accumulation | Learning rate | Epochs |
|---|---|---|---|---|
| Mistral-7B | 4 | 8 | $1.25 \times 10^{-4}$ | 10 |
| LLaMA-3-8B | 4 | 8 | $1.25 \times 10^{-4}$ | 12 |

Table 6: Hyper-parameters for **ablation experiments**.

| BatchSize Per GPU | Gradient Accumulation | Learning rate | Epochs |
|---|---|---|---|
| 4 | 8 | $1.25 \times 10^{-4}$ | 10 |

