# OpenReview forum: "Evo-Step: Evolutionary Generation and Stepwise Validation for Optimizing LLMs in OR"
_ICLR.cc/2025/Conference — Submitted to ICLR 2025_

### Official Review · Reviewer_tXZd · 2024-10-28

**Soundness:** 3
**Presentation:** 3
**Contribution:** 3
**Rating:** 6
**Confidence:** 4

**Summary:**

Evo-Step-Instruct is a framework that uses evolutionary problem generation and stepwise validation to enhance LLMs in Operations Research (OR) tasks. Evo-Step incrementally builds datasets of increasing complexity, applying validation checks to prevent error propagation, and yields strong results on OR benchmarks like NL4OPT and IndustryOR.

**Strengths:**

The evolutionary strategies combined with real-time validation ensure high-quality data generation, reducing the need for post-processing, and the approach significantly outperforms baseline models on complex OR problems.

**Weaknesses:**

The computational demands of the approach could limit scalability, and Evo-Step’s OR-specific design may constrain its applicability to other domains.

**Questions:**

Question 1: Could the authors elaborate on Evo-Step’s computational requirements and its scalability for larger datasets or in different domains?

Question 2: How adaptable is Evo-Step’s framework for domains outside OR, and what modifications would be necessary?

---

> ### Author Response · Authors · 2024-11-21
> **Author respond(1/2)**
>
> We are truly grateful for your careful review and valuable suggestions, which have guided us in strengthening our work. The following responses address your comments in detail.
>
> > Question 1: Could the authors elaborate on Evo-Step’s computational requirements and its scalability for larger datasets or in different domains?
>
> **In the data generation phase**, Evo-Step-Instruct leverages evolution-based generation and stepwise validation mechanisms to create high-quality datasets. The computational requirements in this stage primarily come from iterative problem construction and validation checks.
>
>
> **Efficiency Problem Construction:** The generation mechanism iteratively builds datasets of increasing complexity by reusing solutions from previously generated problems. Instead of re-solving entire problems from scratch, Evo-Step-Instruct focuses only on the newly added or modified components. This significantly reduces computational redundancy and ensures scalability. The validation mechanism focuses on checking specific components (e.g., constraints or objectives) rather than re-evaluating the entire problem, significantly reducing the number of tokens processed per call. These design choices ensure that API usage remains efficient.
>
>
> **Scalability:** Evo-Step-Instruct is designed to scale with dataset size. Its generation mechanism operates iteratively, ensuring that only incremental modifications are made to existing problems rather than constructing them entirely from scratch. The validation overhead is bounded for each individual problem, as the mechanism focuses solely on newly added or modified components. The validation has a fixed limit on the number of iterations. This approach not only minimizes redundant computations but also ensures efficient use of resources, making the framework scalable for large and complex datasets.
>
>
> **In the inference stage**, Evo-Step directly outputs a complete optimization model based on the input problem description. The generated model is then passed to a traditional solver (e.g., COPT, Gurobi) for obtaining solution. Importantly, Evo-Step does not impose additional computational demands during this phase:
>
>
> **Direct Modeling Output:** Evo-Step outputs a fully defined optimization model in one step, requiring neither multi-agent interactions nor iterative reflection to refine solutions. Unlike CoE or Reflexion, which depend on such processes to iteratively improve results, Evo-Step ensures a more efficient inference phase, significantly reducing the computational burden.
>
>
> **No Additional Framework Overhead:** Once the optimization model is generated, the computational effort shifts entirely to the solver. Evo-Step’s role ends at the modeling stage, ensuring that its computational requirements during inference are constant and predictable. Leveraging vLLM for inference, Evo-Step supports parallel generation of multiple optimization models through dynamic batching, allowing efficient processing of concurrent tasks. Once generated, these models can be passed to solvers in parallel, further enhancing scalability and enabling Evo-Step to handle large datasets and high-throughput optimization tasks effectively.

---

> ### Author Response · Authors · 2024-11-21
> **Author respond(2/2)**
>
> > Question 2. How adaptable is Evo-Step’s framework for domains outside OR, and what modifications would be necessary?
>
> Evo-Step-Instruct’s framework is fundamentally adaptable to domains outside OR due to its modular design and reliance on general principles of problem generation and validation.
>
> **Framework adaptability:**
>
> Evo-Step is built on two domain-agnostic principles: iterative Problem Generation and Stepwise Validation Mechanism. The generation mechanism constructs problems by iteratively adding or modifying components, a process that is not tied to OR-specific representations. For example, in WizardLM, a similar strategy is employed to generate instruction-following data by iteratively constructing instructions and responses for LLMs. This demonstrates that Evo-Step-Instruct’s generation approach is well-suited for tasks such as:
>
> 1) Machine learning (hyperparameter optimization or dataset augmentation).
>
> 2) Natural language processing (automated question generation or summarization).
>
> The validation framework ensures the correctness and consistency of generated problems, a concept that generalizes well across domains. For instance:
>
> 1) In natural language processing, validation could verify the semantic coherence of generated instructions.
>
> 2) In machine learning, validation might ensure that generated hyperparameters are within feasible ranges.
>
> These foundational principles make Evo-Step-Instruct inherently adaptable to a wide range of structured problem-solving tasks.
>
> **Necessary Modifications for Other Domains:**
>
> While Evo-Step-Instruct is adaptable, certain domain-specific adjustments are required to maximize its effectiveness: 1) For generation, identifying aspects of the existing problem that can be further evolved, focusing on introducing meaningful complexity or diversity. 2) For validation, specific checks are designed to address different components of the problem, ensuring that all generated aspects meet the necessary requirements and maintain consistency.
>
> **Strengths for adaptation:**
>
> 1) By basing new problem generation on validated existing data, the iterative generation mechanism ensures a baseline level of correctness, avoiding the generation of outlier or nonsensical outputs. This is a key advantage compared to purely generative methods, which often lack such guarantees.
>
> 2) Moreover, the separation of generation and validation processes allows each component to be customized independently, enabling freedom in adapting the framework to new domains without disrupting its core functionality.

---

> > ### Author Response · Authors · 2024-11-29
> > **Looking forward to your reply**
> >
> > Dear Reviewer tXZd,
> >
> > Thank you for your valuable feedback. We have addressed the points you raised in detail and hope that our responses provide further clarification.
> >
> > With the final opportunity for reviewers to post comments approaching on December 2nd, we would greatly appreciate any additional thoughts you may have.
> >
> > Thank you again for your time and consideration.

---

### Official Review · Reviewer_fax4 · 2024-10-28

**Soundness:** 2
**Presentation:** 2
**Contribution:** 2
**Rating:** 5
**Confidence:** 3

**Summary:**

This study proposes Evo-Step-Instruct, a new framework that incrementally escalates problem complexity using an evolutionary approach to strengthen LLMs' performance in optimization modeling. The framework employs progressive validation to enable immediate error identification and correction.

**Strengths:**

Evo-Step exhibits outstanding performance, particularly in managing complex OR tasks, with a significant 17.01% increase in micro-average accuracy on challenging problems.

**Weaknesses:**

The presentation requires substantial improvement, as the current version is difficult to comprehend and does not clearly convey the contributions. Here, I will first list some shortcomings:

(1) The caption of figures is not informed enough and it is difficult to understand the content of diagrams.

(2) Evolutionary Strategy (ES) typically refers to a type of evolutionary algorithm focused on optimizing complex problems by mimicking natural selection processes. However, I cannot see any element of Evolutionary Strategy in Figure 1, which is claimed as the example of evolutionary strategy.

(3) The citation format is not good and needs improvement.

**Questions:**

Due to the presentation, it it hard to fairly evaluate this work at the current stage. As such, I will reconsider this work after the following queries have been addressed:

(1) In this work, I cannot see any evolutionary components such as crossover and mutation. How do you evolve and what do you evolve?

(2) The contribution of this work is not clear or even a bit confused. This work states "increases the complexity of generated problems", why we need to generate problems on our own? A more convincible direction should be that we use LLMs to generate solution to the problems.

(3) How do you fine-tune the open-source LLMs? What technique do you use?

(4) More experiments are required such the the comparison with tradition algorithms (non-LLM approach). LLMs are unnecessarily suited to every task, thus it is important to justify the strength of LLMs in this task.

---

> ### Author Response · Authors · 2024-11-21
> **Author respond(1/3)**
>
> We greatly appreciate your insightful comments and helpful recommendations, which have guided us in improving our work. Our detailed responses to your feedback are provided below.
>
> > Weakness 1. The caption of figures is not informed enough and it is difficult to understand the content of diagrams.
>
> We deeply regret that the captions for our diagrams were not sufficiently detailed in the initial submission. To address this concern, we have revised the captions extensively, providing comprehensive explanations for each figure. These revisions will be incorporated into the updated version of our work, with the aim of improving clarity and enhancing the reader's understanding of the diagrams.
>
> Figure 1 illustrates the Complexity-evolving and Scope-evolving strategies. Complexity-evolving entails methods such as constraint modification, parameter adjustment, and objective alteration, all designed to gradually increase the complexity of problems. Newly introduced or modified components are highlighted in red. In contrast, Scope-evolving emphasizes domain adaptation and problem combination, where problems are transformed to fit new contexts. For example, "theme park" is changed to "hospital," and "scorers" are replaced by "nurses." Problem combinations merge different problem elements to create novel scenarios, and alternative formats, such as tables, are introduced to diversify the representations.
>
> Figure 2 provides a detailed depiction of the Evo-Step-Instruct framework. Starting with an initial dataset, seed data is sampled randomly in each iteration, followed by the use of specific evolving methods by the Problem Generators to create new problem descriptions. These descriptions are evaluated by the Description Checker, which identifies and corrects errors to refine subsequent iterations. Qualified descriptions proceed to the Solution Generators, where solutions are produced and validated through the Variables, Constraints, and Program Checkers. These validation stages ensure the logical consistency and correctness of the solutions. Only solutions passing all checks are incorporated into the dataset for further enhancement. All components—checkers and generators—are implemented using LLMs guided by carefully crafted prompts, with specific elements highlighted in red in the figure.
>
> Figures 3 and 4 highlight key meta-prompts used in the framework.
>
> Figure 3 elaborates on constraint modification, describing stepwise adjustments to constraints that increase problem complexity in a controlled manner.
>
> Figure 4 details the constraint checker prompt, showcasing how it validates the consistency between the problem's constraints and its description, with a particular focus on methods such as Big-M to ensure accurate alignment.
>
> > Weakness 2. Evolutionary Strategy (ES) typically refers to a type of evolutionary algorithm focused on optimizing complex problems by mimicking natural selection processes. However, I cannot see any element of Evolutionary Strategy in Figure 1, which is claimed as an example of evolutionary strategy.
>
> Thank you for pointing out the misunderstanding regarding the use of "evolutionary strategy." We understand that this term is commonly associated with specific algorithms that optimize problems through natural selection-inspired processes. In our work, the term "evolution" is used more generally to describe a step-by-step methodology for generating high-quality data to improve LLM performance in OR problem modeling.
>
> Our framework draws inspiration from WizardLM’s open-domain instruction fine-tuning and adapts it specifically for OR modeling tasks. We introduce two focused categories: Complexity-evolving and Scope-evolving. Both approaches aim to ensure mathematical validity and logical consistency while enriching the diversity of generated problems.
>
> By focusing on these tailored categories, our framework addresses the unique challenges of OR problem modeling without confusion with traditional evolutionary algorithms. We hope this explanation clarifies our methodology and its intended contributions.
>
> > Weakness 3. The citation format is not good and needs improvement.
>
> Thank you for pointing this out. We have thoroughly reviewed the text and identified areas where citation formatting was inconsistent or unclear. For instance, the previous format:
>
> >Operations Research (OR) is a valuable discipline for addressing complex decision-making problems, widely applied in fields such as economics, engineering, and computer science (Bertsimas et al., 2019; Pereira et al., 2022; Belgacem et al., 2020).
>
> Has been revised to:
>
> >Operations Research (OR) is a valuable discipline for addressing complex decision-making problems, widely applied in fields such as economics, engineering, and computer science Bertsimas et al. (2019); Pereira et al. (2022); Belgacem et al. (2020).
>
> These updates will be fully incorporated in the revised submission.

---

> ### Author Response · Authors · 2024-11-21
> **Author respond(2/3)**
>
> > Question 1. In this work, I cannot see any evolutionary components such as crossover and mutation. How do you evolve and what do you evolve?
>
> We would like to clarify the concept of "evolution" as it is applied in Evo-Step-Instruct, as it differs from the traditional evolutionary algorithms that involve genetic operations like crossover and mutation. In Evo-Step-Instruct, "evolution" does not refer to the genetic operations commonly found in evolutionary algorithms. Instead, it denotes a progressive approach to creating a dataset containing OR problems with varying levels of complexity and diversity, specifically for modeling OR problems, rather than optimizing solutions through iterative selection processes.
>
> The Evo-Step framework employs two core evolution-inspired approaches:
>
> Complexity-evolving: This approach iteratively increases the difficulty of problems by adding constraints, modifying parameters, or introducing new objectives.
>
> Scope-evolving: Through domain transitions and problem combinations, this approach broadens the variety of generated problems, thereby expanding capabilities across different scenarios of the dataset.
>
> The evolving methods in Evo-Step are realized through LLMs to directly generate diverse instances, ensuring logical consistency while progressively enhancing the dataset's complexity and scope. This approach focuses on data generation rather than genetic optimization. We hope this clarification provides a better understanding of the Evo-Step-Instruct methodology and look forward to further feedback.
>
> > Question 2. The contribution of this work is not clear or even a bit confused. This work states "increases the complexity of generated problems", why we need to generate problems on our own? A more convincible direction should be that we use LLMs to generate solution to the problems.
>
> We agree that generating solutions with LLMs is a valuable direction. In fact, our work seeks to achieve this goal through an alternative approach by integrating LLMs with solvers, aiming to ultimately improve the overall solution-generation process. Rather than directly using LLMs to produce solutions, we employ LLMs to generate mathematical models and the corresponding programs, which are then used to call solvers to complete the problem-solving process. This method is more reliable, as solvers are specifically designed to handle constraints effectively and ensure the validity of the solutions.
>
>
> The success of generating mathematical models and programs relies heavily on the quality of training data, as such data helps LLMs learn how to accurately structure solving logic and address problem constraints. However, collecting expert-verified data is labor-intensive and requires significant domain expertise, making large-scale collection impractical. To address this challenge, our Evo-Step-Instruct framework generates high-quality datasets based on a small set of verified examples, significantly enriching the training data.
>
>
> By fine-tuning LLMs with this enriched data, our models demonstrate measurable improvements amd achieve superior accuracy compared to GPT-4 with well-designed prompts, particularly for more complex tasks. This enables our framework to effectively address real-world optimization challenges while leveraging the strengths of both LLMs and solvers.
>
>
> We also recognize the need to test this framework across a broader range of optimization tasks and further refine the generated programs for higher generalization, which constitutes a pivotal focus for our future research endeavors.

---

> ### Author Response · Authors · 2024-11-21
> **Author respond(3/3)**
>
> > Question 3. How do you fine-tune the open-source LLMs? What technique do you use?
>
> Details of our fine-tuning process are outlined in Section '4.3 DETAILS,' where we utilize the widely recognized LLaMA-Factory training framework following the Alpaca format template. The input comprises a fixed prompt describing a problem, while the output includes mathematical models and corresponding programs.
>
> We employ the commonly used LoRA fine-tuning technique. For training, we set the learning rate to $1.25\times 10^{-4}$, with 10 and 12 epochs for Mistral-7B and LLaMA-3-8B, respectively. Additional specifics can be found in Appendix A.6.
>
> > Question 4. More experiments are required such the the comparison with tradition algorithms (non-LLM approach). LLMs are unnecessarily suited to every task, thus it is important to justify the strength of LLMs in this task.
>
> The automation of optimization modeling saw significant strides with the introduction of NL4OPT in 2022. Traditional methods like tag-BART, the NeurIPS competition winner, fall under the category of pre-trained language models (PLMs). However, tag-BART lacks code generation capabilities and heavily relies on manual validation to ensure the correctness of constraints and objectives. Despite such manual interventions, its performance on NL4OPT remains at 47.9%, notably lagging behind the results of modern LLM-based approaches, including GPT-4 and ORLM, as demonstrated in Table 11.
>
> **Table 11: Comparative Performance of Various Methods on NL4OPT**
>
> | Category| Method| NL4OPT |
> |-------------------------------|-------------------------|------------|
> |**Methods based on PLMs**|tag-BART|47.9%|
> |**Methods based on GPT-3.5**|Standard| 13.06%|
> ||CoT | 33.06%|
> ||Reflexion| 43.67%|
> ||CoE| 52.24%|
> |**Methods based on GPT-4**|Standard|72.65%|
> ||CoT| 76.73%|
> ||Reflexion|78.78%|
> ||CoE|76.73%|
> |**Fine-tuning LLMs**| ORLM| 78.37%|
> ||Evo-Step-Mistral-7B| 72.65%|
> ||Evo-Step-LLAMA-3-8B| **84.49%**|
>
> These results highlight the limitations of traditional PLM-based approaches like tag-BART, which are unable to achieve satisfactory performance even with extensive manual efforts. In contrast, LLM-based methods leveraging advanced prompt techniques such as CoT and Reflexion exhibit significant advancements, excelling in both code generation and mathematical reasoning to deliver far superior results.
>
> Our proposed framework, Evo-Step-Instruct, further extends these advancements by fine-tuning smaller LLMs (e.g., 7B and 8B parameter models) on high-quality datasets generated through our method. As shown in Table 11, the fine-tuned Evo-Step models outperform other SOTA approaches, including CoE and ORLM, while avoiding reliance on complex prompt engineering or multi-agent frameworks.
>
> This comparison reinforces the limitations of traditional algorithms like tag-BART while illustrating the transformative potential of LLM-based approaches in effectively bridging the gap between natural language problem descriptions and mathematical optimization models.

---

> ### Comment · Reviewer_fax4 · 2024-11-22
>
> Thank you for your response that clarifies the contribution of this work. Importantly, my main concern remains the effectiveness of this overly general framework, since each issue has its own individual characteristics. Misleading terminology that overlaps with the evolutionary computing community also hinders readability, which I think it probably can not be revised at the current stage and needs a thorough revision. I'll therefore keep my current grade.

---

> > ### Author Response · Authors · 2024-11-27
> > **Author respond (3/3)**
> >
> > [1]Xu, Can, et al. "WizardLM: Empowering large pre-trained language models to follow complex instructions." The Twelfth International Conference on Learning Representations. 2024.
> >
> > [2]Tao, Zhengwei, et al. "A survey on self-evolution of large language models." arXiv preprint arXiv:2404.14387 (2024).
> >
> > [3]Soares, Marco Antonio Calijorne, and Fernando Silva Parreiras. "A literature review on question answering techniques, paradigms and systems." Journal of King Saud University-Computer and Information Sciences 32.6 (2020): 635-646.
> >
> > [4]Stahlberg, Felix. "Neural machine translation: A review." Journal of Artificial Intelligence Research 69 (2020): 343-418.
> >
> > [5]El-Kassas, Wafaa S., et al. "Automatic text summarization: A comprehensive survey." Expert systems with applications 165 (2021): 113679.
> >
> > [6]Chowdhery, Aakanksha, et al. "Palm: Scaling language modeling with pathways." Journal of Machine Learning Research 24.240 (2023): 1-113.
> >
> > [7]Achiam, Josh, et al. "Gpt-4 technical report." arXiv preprint arXiv:2303.08774 (2023).
> >
> > [8]Qin, Libo, et al. "Large language models meet nlp: A survey." arXiv preprint arXiv:2405.12819 (2024).
> >
> > [9]Liu, Ze, et al. "Swin transformer: Hierarchical vision transformer using shifted windows." Proceedings of the IEEE/CVF international conference on computer vision. 2021.
> >
> > [10]Ren, Tianhe, et al. "DINO-X: A Unified Vision Model for Open-World Object Detection and Understanding." arXiv preprint arXiv:2411.14347 (2024).
> >
> > [11]Kirillov, Alexander, et al. "Segment anything." Proceedings of the IEEE/CVF International Conference on Computer Vision. 2023.
> >
> > [12]Zhao, Zihan, et al. "ChemDFM: A Large Language Foundation Model for Chemistry." Neurips 2024 Workshop Foundation Models for Science: Progress, Opportunities, and Challenges.
> >
> > [13]Moor, Michael, et al. "Foundation models for generalist medical artificial intelligence." Nature 616.7956 (2023): 259-265.
> >
> > [14]Nguyen, Tung, et al. "ClimaX: A foundation model for weather and climate." arXiv preprint arXiv:2301.10343 (2023).
> >
> > [15]https://deepmind.google/discover/blog/ai-solves-imo-problems-at-silver-medal-level/
> >
> > [16]Liu, Fei, et al. "Multi-task learning for routing problem with cross-problem zero-shot generalization." Proceedings of the 30th ACM SIGKDD Conference on Knowledge Discovery and Data Mining. 2024.
> >
> > [17]Zhou, Jianan, et al. "MVMoE: Multi-Task Vehicle Routing Solver with Mixture-of-Experts." arXiv preprint arXiv:2405.01029 (2024).
> >
> > [18]Anonymous. SHIELD: Multi-task Multi-distribution Vehicle Routing Solver with Sparsity & Hierarchy in Efficiently Layered Decoder. In The Thirteenth International Conference on Learning Representations, 2025. URL: https://openreview.net/forum?id=AMbIvaD4Rr.
> >
> > [19]Berto, Federico, et al. "Routefinder: Towards foundation models for vehicle routing problems." arXiv preprint arXiv:2406.15007 (2024).
> >
> > [20]Wang, Chenguang, and Tianshu Yu. "Efficient training of multi-task neural solver with multi-armed bandits." CoRR (2023).
> >
> > [21]Jiang, Xia, et al. "Unco: Towards unifying neural combinatorial optimization through large language model." arXiv preprint arXiv:2408.12214 (2024).
> >
> > [22]Anonymous. GOAL: A Generalist Combinatorial Optimization Agent Learning. In The Thirteenth International Conference on Learning Representations, 2025. URL: https://openreview.net/forum?id=z2z9suDRjw.
> >
> > [23]Anonymous. Unified Neural Solvers for General TSP and Multiple Combinatorial Optimization Tasks via Problem Reduction and Matrix Encoding. In The Thirteenth International Conference on Learning Representations, 2025. URL: https://openreview.net/forum?id=yEwakMNIex.
> >
> > [24] Xiao, Ziyang, et al. "Chain-of-Experts: When LLMs Meet Complex Operations Research Problems." The Twelfth International Conference on Learning Representations. 2023.
> >
> > [25] Li, Beibin, et al. "Large language models for supply chain optimization." arXiv preprint arXiv:2307.03875 (2023).
> >
> > [26] AhmadiTeshnizi, Ali, Wenzhi Gao, and Madeleine Udell. "OptiMUS: Scalable Optimization Modeling with (MI) LP Solvers and Large Language Models." arXiv preprint arXiv:2402.10172 (2024).
> >
> > [27] Tang, Zhengyang, et al. "ORLM: Training Large Language Models for Optimization Modeling." arXiv preprint arXiv:2405.17743 (2024).
> >
> > [28] Yang, Zhicheng, et al. "Benchmarking llms for optimization modeling and enhancing reasoning via reverse socratic synthesis." arXiv e-prints (2024): arXiv-2407.
> >
> > [29] Ahmed, Tasnim, and Salimur Choudhury. "LM4OPT: Unveiling the potential of Large Language Models in formulating mathematical optimization problems." INFOR: Information Systems and Operational Research (2024): 1-14.
> >
> > [30] Wasserkrug, Segev, Léonard Boussioux, and Wei Sun. "Combining Large Language Models and OR/MS to Make Smarter Decisions." Tutorials in Operations Research: Smarter Decisions for a Better World. INFORMS, 2024. 1-49.

---

> > ### Author Response · Authors · 2024-11-29
> > **Looking forward to your reply**
> >
> > Dear Reviewer fax4,
> >
> > Thank you again for your thoughtful feedback. We have carefully considered your concerns regarding the terminology overlap with the evolutionary computing community. As a result, we have adjusted the manuscript's terminology to address this issue.
> >
> > Regarding your concern about the effectiveness of the general framework, we have provided a detailed response in our previous reply, where we highlight recent advancements not only from other fields, but also within OR, as well as relevant results from both academia and industry. These developments support the feasibility of unified models in addressing a wide range of challenges.
> >
> > As the final opportunity for reviewers to post comments is approaching on December 2nd, we would be grateful for any further thoughts or suggestions you may have. We are more than willing to discuss and address any remaining concerns.
> >
> > Thank you once again for your time and valuable input.

---

> > ### Author Response · Authors · 2024-12-02
> > **Kind Reminder: Reviewer Feedback Deadline**
> >
> > Dear Reviewer fax4,
> >
> > We hope this message finds you well. As the extension period for the review nears its end, we kindly remind you that we have made revisions to the manuscript, including addressing the terminology concerns.
> >
> > In response to your comments on the framework’s effectiveness, we have made further clarifications in the following areas:
> >
> > 1) LLMs' success across domains, with a particular emphasis on mathematics (e.g., AlphaProof for IMO problems).
> >
> > 2) Unified models in machine learning for combinatorial optimization (ML4CO), with Unco utilizing LLMs to solve a variety of problems.
> >
> > 3) The distinction between modeling and solving, highlights that modeling is a simpler task focused on translating problems into formal structures.
> >
> > 4) Empirical validation of automated modeling in both academia and industry (e.g., Chain-of-Experts, OptiGuide).
> >
> > We hope these points clarify your concerns. We believe they collectively demonstrate how LLMs can effectively handle automated modeling for a wide range of combinatorial optimization problems, helping to clarify the framework’s effectiveness.
> >
> > We would be truly grateful if you could revisit your evaluation based on these updates. If there are any further concerns, we would be happy to provide additional information and make every effort to address them.
> >
> > Thank you again for your time and thoughtful consideration.

---

> > ### Author Response · Authors · 2024-12-03
> > **We kindly await your feedback. Thank you for your consideration.**
> >
> > Dear Reviewer fax4,
> >
> > Thank you for your time and effort in reviewing our manuscript and rebuttals.
> >
> > As the author-reviewer discussion is drawing to a close, we hope that our clarifications and revisions have addressed your concerns effectively.  In light of that, we hope you can reconsider your evaluation, taking into account the adjustments we've made. We are open and keen to discuss any remaining concerns and questions you might have.
> >
> > Once again, we sincerely thank you very much.

---

> ### Author Response · Authors · 2024-11-25
> **Author respond (1/3)**
>
> Thank you for your thoughtful feedback
>
> **1) Response to Terminology Concerns**
>
> Regarding the concern about “misleading terminology that overlaps with the evolutionary computing community,” we have carefully revised the manuscript to address this issue. Specifically:
>
> (a) Terms such as “Evolutionary Strategies” have been replaced with “Iterative Problem Generation” to avoid overlap with the terminology of evolutionary computing.
>
> (b) Similarly, “Depth Evolution” and “Breadth Evolution” have been changed to “Complexity-Evolving” and “Scope-Evolving,” respectively.
>
> These adjustments align with terminology commonly used in LLM research, as evidenced by recent works [1, 2]. Therefore, we believe the revised manuscript is more precise and contextually appropriate.
>
> **2) Response to Concerns about Framework Effectiveness**
>
> We understand your concern about the potential limitations of using a general framework for solving distinct problems, especially since each problem has its own unique characteristics. However, recent advancements in machine learning and related fields offer strong evidence supporting the feasibility of unified models tackling diverse challenges effectively.
>
> **(a) Cross-Domain Success of Unified Models**
>
> In NLP, tasks such as Question Answering [3], Machine Translation [4], and Text Summarization [5] each have their own distinct characteristics. Despite these differences, LLMs [6–8] have demonstrated excellent performance across all of these tasks.
>
> This trend extends beyond NLP. In computer vision [9–11], tasks like image classification, segmentation, and object detection, although unique in nature, can all be effectively addressed by unified models. Furthermore, foundational models in fields like chemistry and medicine [12–14] have proven capable of handling a wide range of tasks in their respective domains.
>
> A noteworthy example is Google's AlphaProof [15], which has showcased the ability of unified models to solve specialized problems, such as those from the International Mathematical Olympiad (IMO). AlphaProof's success in solving algebra and number theory problems demonstrates the potential of a single model to handle highly specialized tasks like mathematical proof.
>
>
> **(b) Emerging Trends in Machine Learning for Combinatorial Optimization**
>
> There has been a growing interest in using single models to solve multiple types of combinatorial optimization problems. For example, recent neural solvers have been able to tackle over 10 variants of the Vehicle Routing Problem [16–18], with RouteFinder [19] solving dozens of VRP variants.
>
> Beyond routing problems, frameworks like MAB-MTL [20] have shown promise in addressing diverse problems, including the TSP, the Capacitated Vehicle Routing Problem, the Orienteering Problem (OP), and the Knapsack Problem (KP) with a single model. Unco[21]  utilizes LLMs to extract task-specific features from various COPs and combine them with the initial input features. By leveraging an encoder-decoder architecture, Unco unifies the solving process across multiple problems, such as TSP, CVRP, KP, the Single-Machine Total Weighted Tardiness Problem, and the Minimum Vertex Cover Problem (MVC).
>
> Similarly, GOAL [22] has been trained to solve problems like Asymmetric TSP (ATSP), CVRP, OP, KP, and others, showcasing its impressive ability to generalize across a wide range of combinatorial optimization tasks, including 8 others such as the Traveling Repairman Problem, the Prize Collecting TSP, and others.
>
>  RedCO [23] is trained with ATSP, 2D Euclidean TSP, Directed Hamiltonian Cycle Problem, and Boolean Satisfiability Problem (SAT),  and  prove its ability to generalize across problems like the Vertex Cover Problem, Clique Problem, Independent Set Problem, VRP, MIS, and Flexible Flow Shop Problem (FFSP).
> These examples suggest that solving multiple combinatorial optimization problems with a single model is not only feasible but is increasingly becoming a focal point of research.
>
> **(c) Modeling is a simpler task compared to Solving**
>
> It is important to distinguish between the tasks of modeling and solving in combinatorial optimization. Modeling involves structuring a real-world problem in a formal mathematical way, which is relatively simpler. In contrast, solving the problem requires substantial computational resources and specialized algorithms to find optimal solutions.
>
> Much like machine translation, which translates text from one natural language to another, our task is to translate natural language descriptions into formal mathematical language. This step is easier than solving the problem itself, as it centers on identifying key components—such as variables, constraints, and objectives—and organizing them into a coherent formal structure.
>
> Given the proven ability of LLMs to handle complex mathematical tasks (e.g., AlphaProof’s performance), we believe that a single LLM can effectively address a wide range of combinatorial optimization modeling tasks.

---

> ### Author Response · Authors · 2024-11-25
> **Author respond (2/3)**
>
> **(d) Empirical Validation of Existing Methods**
>
> The task of automated modeling for combinatorial optimization problems has gained significant attention in both the machine learning (ML) and operations research (OR) communities. In the ML field, notable contributions include Chain-of-Experts [24], OptiGuide [25], OptiMUS [26], ORLM [27], and ReSocratic [28]. These works have demonstrated the potential of unified models for modeling a wide range of combinatorial optimization problems. In the OR field, studies such as LM4OPT [29] and others [30] further emphasize the growing interest in this area, showcasing the application of LLMs to solve complex optimization tasks.
>
> In the industrial domain, this research direction has also gained traction. For example:
>
> 1) Microsoft’s OptiGuide [25] optimizes supply chain design for Microsoft Azure, showcasing the real-world application of automated modeling in supply chain optimization.
> 2) Cardinal Operations developed the COLORMind intelligent decision-making platform, based on ORLM [27], which is now being applied in logistics and transportation. (Available at: https://www.shanshu.ai/products/color-mind)
> (Available at: https://www.shanshu.ai/products/color-mind)
> 3) Alibaba’s MindOpt Copilot [https://opt.alibabacloud.com/chat] enables seamless workflows from natural language problem descriptions to modeling and solution generation.
>
> We have validated our proposed model, Evo-Step, using diverse datasets such as NL4Opt, Mamo, and IndustryOR, which cover a wide array of problems including routing, resource allocation, assignment, facility location, investment, etc. The performance of Evo-Step across these diverse problems demonstrates the effectiveness of this approach and further supports the feasibility of using a single model for combinatorial optimization modeling.
>
> Through these empirical validations, we further strengthen the case for automating combinatorial optimization modeling with a single unified model, a concept that has gained both academic and industrial support.
>
> We hope that the clarifications above address your concerns, and we look forward to further discussions.

---

### Official Review · Reviewer_opbT · 2024-11-01

**Soundness:** 2
**Presentation:** 2
**Contribution:** 2
**Rating:** 5
**Confidence:** 4

**Summary:**

This work aims to enhance large language models (LLMs) for complex optimization in Operations Research by evolving problem complexity and integrating real-time validation. Fine-tuned on generated data, the Evo-Step model significantly outperforms benchmarks, improving accuracy on challenging tasks, demonstrating the power of evolutionary problem generation for advanced decision-making automation.

**Strengths:**

1. This work considers different aspects such as problem generation, validation and LLMs fine-tune.
2. The proposed method achieves good performance on different operation problems.

**Weaknesses:**

1. The presentation is poor especially the diagrams. Their captions are so short to understand what they intend to convey. The caption should be significantly expanded and improved.
2. There are some inaccurate expressions. For example, evolutionary strategy specifically refer to a branch algorithm in the evolutionary optimization instead of an arbitrary strategy related to evolution.
3. The problem formulation needs further clarification. Normally, we propose a method/framework to solve a or several problems. However, you are trying to solve a large number of problems simultaneously. How to ensure its effectiveness would be a very serious issue. In addition, the problem to solve is also not clear to readers.
4. I suggest to use the proposed method to solve some OR problems and compare it with some traditional methods instead of LLM-based method, which can help demonstrate the effectiveness of your method.

**Questions:**

1. How do you fine-tune the LLMs? What technique do you use? Details  are lacking.
2. What is the contribution of this work in solving OR problems? It seems to focus on more how to generate problems and validation, which might be more trivival compared to solve problem itself.

---

> ### Author Response · Authors · 2024-11-21
> **Author respond(1/4)**
>
> We deeply value your feedback and thoughtful suggestions, which have significantly helped us refine our work. Each of your comments has been carefully responded to below.
>
> > Weakness 1. The presentation is poor especially the diagrams. Their captions are so short to understand what they intend to convey. The caption should be significantly expanded and improved.
>
> We sincerely apologize for not providing sufficiently detailed captions for the diagrams. To address this, we have prepared expanded captions with detailed explanations for each image, which will be incorporated into the revised version to improve readability and ensure that each diagram is more informative for the reader.
>
> Figure 1 demonstrates Complexity-evolving and Scope-evolving approaches. Complexity-evolving includes constraint modification, parameter adjustment, and objective alteration, which incrementally increase problem difficulty. Red-highlighted text represents newly added or modified elements. Conversely, Scope-evolving focuses on domain transformation and problem combination, adapting problems to different contexts. For instance, "theme park" is replaced with "hospital," and "scorers" becomes "nurses." Combination integrates elements from distinct problems, creating entirely new scenarios. Additionally, new formats such as tables are introduced.
>
> To provide clarity on the implementation of these elements, Figure 2 illustrates the Evo-Step-Instruct framework. Given an initial dataset, each iteration involves randomly sampling seed data, after which a specific evolving approach is used by the Problem Generators to create a new problem description. This description is checked by the Description Checker, which provides feedback on detected errors to guide the next round of generation. When the description is qualified, Solution Generators produce a solution. The solution is then validated by the Variables, Constraints, and Program Checkers. Errors are also provided as feedback to help resolve issues. Only solutions that pass all checks are considered qualified. These solutions, along with their descriptions, are added to the dataset for further generation. All the checkers and generators are implemented using LLMs with well-designed prompts, as shown in the figure. The red words highlight special requests of different prompts.
>
>
> Figures 3 and 4 provide meta-prompts for key components of the framework.
>
> Figure 3 focuses on constraint modification, detailing how to adjust constraints incrementally to increase complexity.
>
> Figure 4 outlines the constraint checker prompt, illustrating how it ensures alignment between model constraints and problem descriptions, with particular attention to techniques like the Big-M method.
>
> We hope these expanded captions will assist readers in gaining a clearer understanding of the key aspects of our work.
>
> > Weakness 2. There are some inaccurate expressions. For example, evolutionary strategy specifically refer to a branch algorithm in the evolutionary optimization instead of an arbitrary strategy related to evolution.
>
> Thank you for pointing out the inaccurate expression of evolutionary strategy. To clarify, our approach begins with a simple OR problem and gradually increases its complexity and diversity in a step-by-step manner.The concept of evolution in our framework is inspired by WizardLM, which focuses on tasks related to open-domain Instruction fine-tuning. However, our work specifically adapts this idea to the context of OR problem modeling, introducing two tailored categories: Complexity-evolving and Scope-evolving.
>
> By explicitly focusing on these categories, our framework addresses the unique requirements of OR problems, including the generation of mathematically valid constraints and problem formulations, while avoiding confusion with the term "evolutionary algorithms."

---

> ### Author Response · Authors · 2024-11-21
> **Author respond(2/4)**
>
> > Weakness 3. The problem formulation needs further clarification. Normally, we propose a method/framework to solve a or several problems. However, you are trying to solve a large number of problems simultaneously. How to ensure its effectiveness would be a very serious issue. In addition, the problem to solve is also not clear to readers.
>
> We sincerely appreciate your thoughtful observations about the scope of our work and its relation to the broader field of combinatorial optimization. It is true that most existing studies primarily focus on applying a single method or framework to solve one or a few specific types of combinatorial optimization problems [1,2,3]. However, with the advancement of machine learning models' generalization capabilities, recent research, such as GOAL [4], has begun to explore the potential of using a single model to address a variety of distinct problem types.
>
> Large language models (LLMs), known for their strong generalization abilities [5], hold significant potential for addressing a wide range of problem types. Prior studies, including CoE, Optimus, and ORLM, have already demonstrated the capability of LLMs to address various combinatorial optimization problems, such as planning, routing, allocation, and assignment. These approaches do not directly provide solutions but instead formularize natural language problem descriptions into mathematical models and solver-compatible programs, which are subsequently solved using solvers (like Gurobi and COPT). Notably, CoE and Optimus have shown that LLMs possess inherent modeling capabilities for diverse problems through pre-training, while ORLM has enhanced this ability by generating high-quality data for further fine-tuning.
>
> Building on these advancements, our work leverages the Evo-Step-Instruct framework to generate datasets that further improve LLMs’ capabilities, particularly in terms of generalization. The framework emphasizes three critical aspects:
>
> 1) enhancing the diversity of the training dataset to better capture a wide range of problem scenarios.
>
> 2) ensuring the dataset includes examples spanning various levels of difficulty.
>
> 3) maintaining data quality through precise problem descriptions and accurate modeling.
>
> Our framework specifically emphasizes diversity by generating problems of varying difficulty levels and from different domains, ensuring that fine-tuned models can effectively address a broader spectrum of optimization tasks.
>
> The experimental results demonstrate that Evo-Step-Instruct significantly enhances model performance across various benchmarks, such as NL4OPT, MAMO, and industry, which include problems of different types and complexities.
>
> In summary, by leveraging the inherent generalization capabilities of LLMs and emphasizing diversity and quality in data generation, our Evo-Step-Instruct framework ensures both the effectiveness and adaptability of LLMs in modeling a wide range of OR problems. This targeted effort addresses concerns about their capability to handle multiple problem types while significantly enhancing their performance and robustness.
>
>
> [1] Bengio, Yoshua, Andrea Lodi, and Antoine Prouvost. "Machine learning for combinatorial optimization: a methodological tour d’horizon." European Journal of Operational Research 290.2 (2021): 405-421.
>
> [2] Cappart, Quentin, et al. "Combinatorial optimization and reasoning with graph neural networks." Journal of Machine Learning Research 24.130 (2023): 1-61.
>
> [3] Mazyavkina, Nina, et al. "Reinforcement learning for combinatorial optimization: A survey." Computers & Operations Research 134 (2021): 105400.
>
> [4] Drakulic, Darko, Sofia Michel, and Jean-Marc Andreoli. "GOAL: A Generalist Combinatorial Optimization Agent Learning." arXiv preprint arXiv:2406.15079 (2024).
>
> [5] Yang, Haoran, et al. "Unveiling the Generalization Power of Fine-Tuned Large Language Models." Proceedings of the 2024 Conference of the North American Chapter of the Association for Computational Linguistics: Human Language Technologies (Volume 1: Long Papers). 2024.

---

> ### Author Response · Authors · 2024-11-21
> **Author respond(3/4)**
>
> > Weakness 4. I suggest to use the proposed method to solve some OR problems and compare it with some traditional methods instead of LLM-based method, which can help demonstrate the effectiveness of your method.
>
> The progress of automating optimization modeling began with the competition of NL4OPT in 2022. Traditional methods, such as the tag-BART model that won the NeurIPS competition, belong to pre-trained language models (PLMs) but lack code generation capabilities and require extensive manual validation of constraints and objectives to ensure correctness. Despite these manual interventions, tag-BART's performance on NL4OPT is 47.9%, a result that falls significantly behind the performance of LLM-based methods, including GPT-4 and ORLM, as shown in Table 10.
>
> **Table 10: Comparative Performance of Various Methods on NL4OPT**
> | Category| Method|NL4OPT|
> |-------------------------------|-------------------------|------------|
> | **Methods based on PLMs**| tag-BART| 47.9%|
> | **Methods based on GPT-3.5**  | Standard| 13.06%|
> || CoT | 33.06%|
> || Reflexion| 43.67%|
> ||CoE| 52.24%|
> |**Methods based on GPT-4**|Standard| 72.65%|
> ||CoT| 76.73%|
> ||Reflexion|78.78%|
> ||CoE| 76.73%|
> |**Fine-tuning LLMs** |ORLM|78.37%|
> ||Evo-Step-Mistral-7B| 72.65%|
> ||Evo-Step-LLAMA-3-8B|**84.49%**|
>
> The data underscores that PLM-based models, such as tag-BART, struggle to deliver adequate performance in optimization modeling tasks, even with manual intervention. In contrast, LLM-based methods, particularly those leveraging advanced techniques such as CoT and Reflexion, have made substantial progress in automating both code generation and mathematical reasoning, achieving far superior performance.
>
> Our proposed method further builds on these advancements by fine-tuning smaller LLMs (e.g., with 7B or 8B parameters) using high-quality data generated by Evo-Step-Instruct framework. As demonstrated in the table, our fine-tuned Evo-Step models surpass other SOTA methods, such as CoE and ORLM, without relying on well-designed prompts or multi-agent frameworks.
>
> This performance comparison underscores the limitations of traditional methods such as tag-BART and highlights the promise of advanced LLM-based techniques in bridging the gap between natural language descriptions and mathematical models.
>
> > Question 1. How do you fine-tune the LLMs? What technique do you use? Details are lacking.
>
> Details on our fine-tuning process are provided in section '4.3 DETAILS.', where we describe the use of the widely used LLaMA-Factory training framework, utilizing the Alpaca format template. In this setup, the input consists of a fixed prompt with a problem description, and the output is a solution that includes mathematical models and corresponding programs. For specific technique, we use the usually adopted fine-tune method LoRA .
>
> For training, we set the learning rate as $1.25\times 10^{-4}$, and epoches are 10 and 12 for Mistral-7B and LLaMA-3-8B, respectively. The details of training are listed in Appendix A.6.

---

> ### Author Response · Authors · 2024-11-21
> **Author respond(4/4)**
>
> > Question 2. What is the contribution of this work in solving OR problems? It seems to focus on more how to generate problems and validation, which might be more trivival compared to solve problem itself.
>
> Our work contributes to propose a novel high quality training data generation method to train LLMs to have the capacity to formularize real-world problems described in natural language into mathematical models and corresponding solution programs. This step is a significant part of solving real-world OR problems. Only after these problems are transformed into mathematical models can they be solved using OR techniques, including solvers. According to a report from Gurobi[1], 81% of Gurobi solver users hold advanced degrees, and 49% of all respondents major in OR. Such an expertise gap limits the usefulness of operations research in a broader range of practical applications.
>
>
> Given this expertise gap, we do not view the tasks of problem generation and validation for LLM training as trivial compared to problem-solving. In fact, the current capabilities of LLMs are inadequate for fully automating the optimization modeling of real-world OR problems. Enhancing their modeling capabilities is a substantial challenge that demands urgent attention.
>
>
> A critical aspect of improving LLMs' modeling capabilities lies in **the quality of training data**, which fundamentally determines model performance. However, existing fine-tuning datasets revealed a significant number of errors, which can negatively impact model performance if used for fine-tuning. Moreover, previous datasets often lack sufficient diversity, which limits the generalization capability of models to handle a wide range of OR tasks.
>
>
> To address these challenges, we propose a novel data augmentation framework based on evolutionary methods. During the generation process, we introduce Complexity-evolving to produce data by increasing complexity and refining problem constraints. Additionally, Scope-evolving generates diverse variations of existing problems to cover a broader range of scenarios. To ensure the accuracy of the generated data, we apply stepwise validation mechanism to filter and validate the quality of the augmented dataset.
>
>
> By enhancing the complexity and diversity of fine-tuning data, our framework enables LLMs to better tackle optimization modeling. The enhanced datasets significantly improve the modeling capabilities of LLMs, making a substantial contribution toward bridging the gap between natural language problem descriptions and effective OR problem-solving.
>
> [1] Gurobi Optimization. matical optimization report, 2023. URL https://www.gurobi.com/resources/report-state-of-mathematical-optimization-2023/.

---

> > ### Comment · Reviewer_opbT · 2024-11-26
> >
> > I appreciate your detailed response. The comparative results are all based on LLMs while I would like to see the comparison with traditional methods (non LLM-based). Such comparisons might be necessary to make the findings convincing. Thus, it is unclear what the proposed methods can solve specific problems yet. I would maintain my initial score.

---

> > > ### Author Response · Authors · 2024-11-27
> > > **Author respond(1/2)**
> > >
> > > Our work centers on achieving **automated modeling for combinatorial optimization problems.** Specifically, we aim to **take a natural language description of a combinatorial optimization problem as input and produce a corresponding mathematical model and solution program as output.** This program can directly call solvers to obtain optimal solutions. It is important to clarify that **our goal is not to solve already modeled optimization problems but to automate the modeling process itself.** Below is a specific example that demonstrates the input and output for an instance:
> > >
> > > **Example:**
> > >
> > > **Input:** A logistics company wants to allocate up to 250 vehicles between operations A and B. To ensure efficiency, the difference between twice the vehicles in A and those in B must be at least 60 units. Additionally, The number of vehicles in A plus twice the number in B should not exceed 300 units. Each vehicle costs 400 for A and 300 for B. What would be the minimum total cost (in dollars)?
> > >
> > > **Output:**
> > >
> > > Mathematical Model:
> > >
> > > Variables:
> > > - $x_A, x_B$: Non-negative integer variables representing the number of vehicles allocated to A and B, respectively.
> > > - $z$: A binary variable to model the efficiency constraint.
> > >
> > > Objective Function:
> > >
> > > Minimize:  $Z = 400 \cdot x_A + 300 \cdot x_B$
> > >
> > >
> > > Constraints:
> > > 1. Total number of vehicles constraint:
> > >
> > > $$x_A + x_B \leq 250$$
> > >
> > > 2. Efficiency condition constraint (Introducing a sufficiently large constant $M$):
> > >
> > > $ 2x_A - x_B \geq 60 - M \cdot z$
> > >
> > > $x_B - 2x_A \geq 60 - M \cdot (1 - z)$
> > >
> > > 3. Capacity condition:
> > >
> > > $x_A + 2x_B \leq 300$
> > >
> > >
> > > ```python
> > > import coptpy as cp
> > > from coptpy import COPT
> > >
> > > # Create an environment and model
> > > env = cp.Envr()
> > > model = env.createModel("LogisticsVehicleAllocation")
> > >
> > > # Add variables
> > > x_A = model.addVar(vtype=COPT.INTEGER, name="x_A")
> > > x_B = model.addVar(vtype=COPT.INTEGER, name="x_B")
> > > z = model.addVar(vtype=COPT.BINARY, name="z")
> > >
> > > # Set the objective function
> > > model.setObjective(400 * x_A + 300 * x_B, sense=COPT.MINIMIZE)
> > >
> > > # Add constraints
> > > model.addConstr(x_A + x_B <= 250, name="TotalVehicles")
> > > model.addConstr(2 * x_A - x_B >= 60 - 1000 * z, name="EfficiencyConstraint1")
> > > model.addConstr(x_B - 2 * x_A >= 60 - 1000 * (1 - z), name="EfficiencyConstraint2")
> > > model.addConstr(x_A + 2 * x_B <= 300, name="CapacityCondition")
> > >
> > > # Solve the model
> > > model.solve()
> > >
> > > # Output results
> > > if model.status == COPT.OPTIMAL:
> > >     print(f"x_A: {x_A.x}")
> > >     print(f"x_B: {x_B.x}")
> > >     print(f"Objective Value: {model.objVal}")
> > > ```
> > >
> > > Automated modeling for combinatorial optimization problems is a relatively new research area that began gaining traction in 2022, largely driven by the introduction of the NL4Opt competition[1]. This competition introduced 1,101 linear programming problems, with 289 designated for testing, and was designed to reduce reliance on OR professionals and manual design. Historically, modeling tasks depended heavily on significant domain expertise, limiting accessibility. The NL4Opt competition addressed this by promoting ML approaches to automate modeling.
> > >
> > > The NL4Opt competition divided the modeling task into two subtasks:
> > >
> > > 1) Named Entity Recognition (NER): Identifying and labeling semantic entities corresponding to the components of an optimization problem.
> > >
> > > 2) Generation: Producing a logical form of the problem based on these identified entities.
> > >
> > > In the competition, the baseline model employed a BART[2] encoder-decoder architecture, and the top five submissions were built upon pre-trained language models (PLMs) such as BART or T5[3], rather than LLMs. These methods utilized targeted designs, including input tagging, adversarial training, and multitask learning, to address the task. However, their performance revealed significant limitations in both generalization and scalability. For instance, tag-BART[4], which won 1st place, achieved only 47.9% accuracy on NL4OPT , far below the results of GPT-4 or fine-tuned LLMs, as detailed in Table 10: Comparative Performance of Various Methods on NL4OPT. Moreover, tag-BART required manual inspection of output equations for correctness, underscoring its reliance on human intervention.
> > >
> > > In fact, generating formalized mathematical models and executable code from natural language descriptions is inherently suited to LLMs, which excel in semantic understanding and logical generation. **Even the advanced PLMs also face challenges in effectively handling this task. Traditional non-LLM methods struggle even more, making meaningful comparisons across these approaches inherently difficult for this specific domain.**

---

> > > ### Author Response · Authors · 2024-11-27
> > > **Author respond(2/2)**
> > >
> > > The task of automated modeling for combinatorial optimization problems gains significant attention in both the ML and operations research (OR) communities. Notable contributions in the ML field include Chain-of-Experts[5], OptiGuide[6], OptiMUS[7], ORLM[8], and ReSocratic[9], along with works such as CAFA [10]. In the OR field, studies like LM4OPT [11] and others [12] further highlight the growing interest in this area. It is worth noting that all of these efforts have relied exclusively on LLMs.
> > >
> > > In the industrial domain, this research direction has also gained traction. For example:
> > > 1) Microsoft’s OptiGuide[6] optimizes supply chains for Microsoft Azure.
> > >
> > > 2) Cardinal Operations developed the COLORMind intelligent decision-making platform[https://www.shanshu.ai/products/color-mind](https://www.shanshu.ai/products/color-mind) based on ORLM[8], now applied in logistics and transportation.
> > >
> > > 3) Alibaba’s MindOpt Copilot [https://opt.alibabacloud.com/chat] (https://opt.alibabacloud.com/chat) enables seamless workflows from natural language problem descriptions to modeling and solution generation.
> > >
> > > All these industrial applications leverage the capabilities of LLMs, further demonstrating their pivotal role in this domain.
> > > Both academia and industry have increasingly focused on automated modeling for combinatorial optimization problems, driven by real-world needs and the demonstrated effectiveness of LLM-based approaches. **Except for the earliest methods developed for the NL4Opt competition [4,13,14,15], which relied on PLMs, all subsequent advancements have been exclusively based on LLMs.** This clearly demonstrates the fundamental suitability of LLMs for addressing this challenging task.
> > >
> > > Building on this context and following the trajectory of prior work, our approach aligns with the current research landscape by conducting comprehensive comparisons with LLM-based methods, which represent the state-of-the-art in automated modeling. Furthermore, we benchmark our method against the representative PLM-based approach, tag-BART, to effectively highlight the advancements achieved in this domain.
> > >
> > > [1] Ramamonjison, Rindranirina, et al. "Nl4opt competition: Formulating optimization problems based on their natural language descriptions." NeurIPS 2022 Competition Track. PMLR, 2023.
> > >
> > > [2] Lewis, M. "Bart: Denoising sequence-to-sequence pre-training for natural language generation, translation, and comprehension." arXiv preprint arXiv:1910.13461 (2019).
> > >
> > > [3] Raffel, Colin, et al. "Exploring the limits of transfer learning with a unified text-to-text transformer." Journal of machine learning research 21.140 (2020): 1-67.
> > >
> > > [4] Gangwar, Neeraj, and Nickvash Kani. "Highlighting named entities in input for auto-formulation of optimization problems." International Conference on Intelligent Computer Mathematics. Cham: Springer Nature Switzerland, 2023.
> > >
> > > [5] Xiao, Ziyang, et al. "Chain-of-Experts: When LLMs Meet Complex Operations Research Problems." The Twelfth International Conference on Learning Representations. 2023.
> > >
> > > [6] Li, Beibin, et al. "Large language models for supply chain optimization." arXiv preprint arXiv:2307.03875 (2023).
> > >
> > > [7] AhmadiTeshnizi, Ali, Wenzhi Gao, and Madeleine Udell. "OptiMUS: Scalable Optimization Modeling with (MI) LP Solvers and Large Language Models." arXiv preprint arXiv:2402.10172 (2024).
> > >
> > > [8] Tang, Zhengyang, et al. "ORLM: Training Large Language Models for Optimization Modeling." arXiv preprint arXiv:2405.17743 (2024).
> > >
> > > [9] Yang, Zhicheng, et al. "Benchmarking llms for optimization modeling and enhancing reasoning via reverse socratic synthesis." arXiv e-prints (2024): arXiv-2407.
> > >
> > > [10] Deng, Haoxuan, et al. "CAFA: Coding as Auto-Formulation Can Boost Large Language Models in Solving Linear Programming Problem." The 4th Workshop on Mathematical Reasoning and AI at NeurIPS'24.
> > >
> > > [11] Ahmed, Tasnim, and Salimur Choudhury. "LM4OPT: Unveiling the potential of Large Language Models in formulating mathematical optimization problems." INFOR: Information Systems and Operational Research (2024): 1-14.
> > >
> > > [12] Wasserkrug, Segev, Léonard Boussioux, and Wei Sun. "Combining Large Language Models and OR/MS to Make Smarter Decisions." Tutorials in Operations Research: Smarter Decisions for a Better World. INFORMS, 2024. 1-49.
> > >
> > > [13]Jang, Sanghwan. "Tag embedding and well-defined intermediate representation improve auto-formulation of problem description." arXiv preprint arXiv:2212.03575 (2022).
> > >
> > > [14]Ning, Yuting, et al. "A novel approach for auto-formulation of optimization problems." arXiv preprint arXiv:2302.04643 (2023).
> > >
> > > [15]He, JiangLong, et al. "Linear programming word problems formulation using ensemblecrf ner labeler and t5 text generator with data augmentations." arXiv preprint arXiv:2212.14657 (2022).

---

> > > ### Author Response · Authors · 2024-11-29
> > > **Looking forward to your reply**
> > >
> > > Dear Reviewer opbT,
> > >
> > > Thank you again for your valuable feedback. We have fully addressed your concerns regarding the comparison with traditional methods, and we hope the explanations provided are clear.
> > >
> > > We eagerly anticipate any further suggestions or thoughts you may have, as the final opportunity for reviewers to post comments is approaching on December 2nd. We are more than happy to discuss and resolve any remaining concerns.
> > >
> > > Thank you for your time and consideration.

---

> > > > ### Comment · Reviewer_opbT · 2024-11-30
> > > >
> > > > I appreciate your effort and detailed response. I have re-evaluated this work and will adjust my score accordingly.

---

> > > > > ### Author Response · Authors · 2024-12-01
> > > > > **Author respond**
> > > > >
> > > > > Dear Reviewer opbT,
> > > > >
> > > > > Thank you very much for your kind acknowledgment of our efforts and responses. We are deeply grateful for the time and thought you have invested in reviewing our work. In response to your valuable suggestions, we have carefully revised the manuscript, particularly by expanding the figure captions, which we believe has greatly enhanced the clarity and readability of the paper.
> > > > >
> > > > > While we sincerely appreciate the score adjustment, we noticed that no additional concerns were explicitly mentioned in your feedback. If possible, we would be extremely grateful if you could kindly share any remaining issues or areas of concern.
> > > > >
> > > > > Your thoughtful feedback is of utmost importance to us, as it provides an invaluable opportunity to improve the quality of our work and manuscript. We are eager to learn from your insights and will make every effort to address any remaining concerns.
> > > > >
> > > > > Thank you once again for your time and consideration.

---

### Official Review · Reviewer_Je8b · 2024-11-03

**Soundness:** 3
**Presentation:** 3
**Contribution:** 3
**Rating:** 6
**Confidence:** 4

**Summary:**

This paper proposes an evolutionary framework for data generation for training LLMs for OR optimization modelling. The data generation consists of two parts, with the first evolutionary one, for instance, generation and the second one for solution assessment and refinement. The results are compared on three OR datasets of different difficulty. Promising results are generated when compared to existing LLMs with prompt engineering techniques and domain OR LLMs.

**Strengths:**

1.The framework automates the data generation process, enhancing efficiency and effectiveness.
2. The paper is well-structured and easy to follow, facilitating comprehension.

**Weaknesses:**

1. Although the results contribute to specific OR optimization problem modeling, the technique's contribution appears incremental, as the evolution process in data generation has been previously explored.
2. It is recommended to include more comparative studies with SOTA LLMs and provide additional explanations to further validate the results.

**Questions:**

1. How many queries are utilized during instance generation? What is the total cost? What percentage of samples are discarded during the data generation process?
2. The authors mention testing both COPT and GUROBI. It is unclear whether they use the same independent data generation pipeline and prompts for each.
3. Are all experimental results based on COPT, GUROBI, or are they selecting the best outcomes from either? Additionally, if OMLR is only trained with COPT, is it appropriate to use the original checkpoint directly?
4. What would be the impact of directly applying the proposed "Stepwise Validation Mechanism" to SOTA LLMs, such as GPT-4, in modeling tasks? Would this significantly enhance accuracy and performance?

---

> ### Author Response · Authors · 2024-11-21
> **Author respond(1/3)**
>
> Thank you for your insightful feedback and constructive suggestions, which have been invaluable for improving our work. Below, we provide detailed responses to your comments.
>
>
> > Weakness 1. Although the results contribute to specific OR optimization problem modeling, the technique's contribution appears incremental, as the evolution process in data generation has been previously explored.
>
> We acknowledge that the evolution process in data generation has been explored in prior works. However, we believe our contribution addresses critical gaps in these methods, particularly in the context of OR problem modeling.
>
> **Challenges in OR Data Generation**
>
> Generating high-quality datasets for OR modeling presents unique challenges that general data generation approaches fail to address:
>
> 1) Constraint Feasibility: OR problems require mathematically valid and solvable models, where variables, constraints and objectives must align. This is not considered by general-purpose data generation methods.
>
> 2) Diversity in Problem Complexity: High-quality datasets should include problems spanning varying difficulty levels to enable comprehensive model training. However, existing methods, including ORLM’s prompt-based generation, lack the ability to control problem complexity, resulting in datasets dominated by either overly simple or overly complex problems.
> 3) Validation Beyond Syntax: Existing generation usually adopt rule-based postprocessing methods, which can filter out formatting and syntax errors, but is inapplicable of identifying deeper logical or mathematical inconsistencies, which are critical for real-world OR applications.
>
> **Our Contributions**
>
> To overcome these challenges, our method introduces two key innovations that enhance the quality and applicability of OR datasets:
>
> 1) Iterative Problem Generation: We implement a progressive generation strategy, where problems are generated with gradually increasing complexity. This ensures a balanced dataset with problems of varying difficulty, from simple to complex. Such a approach allows us to enrich the training corpus, enabling fine-tuned LLMs to handle diverse problem formulations and adapt to a wide range of real-world scenarios.
>
> 2) Stepwise Validation Mechanism: Our framework incorporates a modular validation process designed to ensure that generated problems are not only free from syntax errors but also devoid of deeper logical and modeling inconsistencies. It verifies logical consistency by checking the coherence between constraints, objectives, and variables, ensures solver compatibility through mathematical feasibility checks, and aligns generated problems with real-world OR scenarios. Additionally, the mechanism validates the accompanying code for compatibility with solvers like Gurobi or COPT, eliminating runtime errors and structural inconsistencies. These comprehensive checks guarantee that the datasets are accurate, reliable, and applicable to practical optimization tasks.
>
> Our framework demonstrates significant advancements over SOTA methods across multiple benchmarks. The results highlight Evo-Step-Instruct’s ability to generate high-quality datasets that enhance LLM performance in OR modeling tasks. By enabling LLMs to more effectively translate natural language problem descriptions into accurate mathematical models, we believe this targeted effort has resulted in meaningful progress towards addressing key challenges in OR problem modeling.

---

> ### Author Response · Authors · 2024-11-21
> **Author respond(2/3)**
>
> > Weakness 2. It is recommended to include more comparative studies with SOTA LLMs and provide additional explanations to further validate the results.
>
> Thank you for your thoughtful suggestion. To further validate our results, we conducted comparative studies on the MAMO ComplexLP dataset, involving leading proprietary LLM GPT-4o-2024-08-06 and an advanced open-source LLM Qwen2.5-72B-Instruct. These comparisons provide additional context to the effectiveness of our Evo-Step framework. The results are summarized in Table 7 below:
>
> **Table 7: Performance Comparison of Various Methods on MAMO ComplexLP**
> | **Model\Method**              | Standard| CoT | Reflexion| CoE  | Accuracy |
> |-------------------------------|--------------|----------|---------------|----------|------------------|
> | GPT-3.5| 10.90%       | 13.27%   | 14.22%        | 17.06%   |                  |
> | GPT-4 | 24.64%       | 29.86%   | 36.02%        | 40.28%   |                  |
> | GPT-4o        | 46.92%       | 49.29%   | 48.34%        | 54.03%|                  |
> | Qwen2.5-72B-Instruct    | 46.45%       | 45.97%   | 47.87%        | 51.66%|                  |
> | ORLM      | -            | -        | -             | -        | 38.39%       |
> | Evo-Step-Mistral-7B | -            | -        | -             | -        | 52.61%      |
> | Evo-Step-LLaMA-3-8B      | -            | -        | -             | -        | 61.61%      |
>
> The results on the MAMO ComplexLP dataset highlight the advancements of both proprietary and open-source LLMs in OR modeling tasks. Proprietary models like GPT-4o demonstrate notable improvements, achieving a maximum accuracy of 54.03% with CoE and consistently outperforming earlier versions like GPT-4 and GPT-3.5. Similarly, open-source models such as Qwen2.5 achieve competitive results, with Reflexion reaching 47.87% and CoE achieving 51.66%. These findings indicate that open-source models are steadily narrowing the gap with proprietary counterparts, even without task-specific fine-tuning.
>
> Despite these advancements, Evo-Step still demonstrates significant superiority, achieving the highest accuracy of 61.61% with Evo-Step-LLaMA-3-8B, surpassing GPT-4o and other baselines. Evo-Step-Mistral-7B also achieves 52.61%, further showcasing the effectiveness of our framework. These results emphasize the impact of Evo-Step’s task-specific training data in elevating model performance across diverse problem formulations.
>
> By generating high-quality, diverse datasets, Evo-Step addresses a key challenge in structured optimization tasks: enabling LLMs to better handle complex problems. The consistent performance of Evo-Step-trained models highlights the importance of integrating precise, task-specific data into fine-tuning pipelines, paving the way for more reliable and effective solutions to real-world optimization challenges.
>
> >Question 1. How many queries are utilized during instance generation? What is the total cost? What percentage of samples are discarded during the data generation process?
>
> The instance generation involved 64K queries, and the number of tokens was 179M. On average, each generation iteration required approximately 7.66 queries, with 3.14 queries dedicated to generating and validating the problem description, and 4.52 queries used for solution generation and validation. Of the total tokens, 39M were allocated to generating and validating the problem description, while the remaining 140M were used for solution generation and validation. Additionally, 8,400 generations were conducted, yielding 4,464 samples, meaning that 46.86% of the generated samples were discarded.
>
> >Question 2. The authors mention testing both COPT and GUROBI. It is unclear whether they use the same independent data generation pipeline and prompts for each.
>
> In Section 4.2 Baseline, we have already mentioned that for prompt engineering methods, we tested results separately using both Gurobi and COPT, reporting only the best-performing result for each method.
>
> For fine-tuning methods, we exclusively used the COPT solver. This choice was guided by two considerations:
>
> 1) The raw data collected for our experiments was based on COPT, making it the natural basis for all derived datasets in fine-tuning experiments.
>
> 2) Using COPT ensures a fair comparison with the ORLM method, which also adopts COPT as its solver.
>
> To further clarify this in the manuscript, we plan to revise the Section 4.2 in the manuscript to explicitly state:
>
> "Prompt engineering methods were evaluated independently using both COPT and Gurobi solvers, with the best result reported. Fine-tuning methods, including Evo-Step, exclusively utilized the COPT solver due to the alignment of collected raw data and to ensure a fair comparison with ORLM, which also employs COPT."
>
> We hope this clarification resolves your concern.

---

> ### Author Response · Authors · 2024-11-21
> **Author respond(3/3)**
>
> >Question 3. Are all experimental results based on COPT, GUROBI, or are they selecting the best outcomes from either? Additionally, if OMLR is only trained with COPT, is it appropriate to use the original checkpoint directly?
>
> We appreciate the your thoughtful questions and would like to provide additional clarification.
>
> (1) For the prompt engineering methods, we evaluated each method independently using both COPT and Gurobi and reported the best-performing result. For the fine-tuning methods, including Evo-Step and ORLM, all results are based solely on the COPT, as the raw data used for fine-tuning was generated using COPT.
>
> (2) The original ORLM checkpoint was used without modification to ensure fairness and consistency. Since the authors of ORLM only released the checkpoint and 3,000 training samples (rather than the full dataset), it would be impossible to reproduce their results or achieve a fair comparison by re-training or converting their data to Gurobi. Using the checkpoint maintains methodological consistency with the original work, which was exclusively fine-tuned on COPT data, ensuring a fair comparison.
>
> (3) To validate the necessity of fine-tuning and the contribution of our data generation framework, we tested LLaMA-3-8B and Mistral-7B directly (without fine-tuning) on the NL4OPT, MAMO and IndustryOR datasets. The results, shown in Table 8, indicate that both models achieved 0% accuracy across all benchmarks, highlighting that these models alone are insufficient for modeling OR problems. This underscores the importance of high-quality datasets and the advantage provided by our data generation framework.
>
> In contrast, for the prompt engineering methods, larger models such as GPT were used, leveraging their embedded knowledge along with carefully designed prompts. To minimize biases from solver-specific knowledge and ensure fairness, these methods were evaluated separately using both COPT and Gurobi solvers.
>
> **Table 8: Performance Comparison of Base and Fine-Tuned Models on Different Datasets**
>
> | **Model**          | NL4OPT | MAMO EasyLP | MAMO ComplexLP | IndustryOR |
> |---------------------|------------|------------------|---------------------|-----------------|
> | Mistral-7B    | 0%  | 0%  | 0%                  | 0%              |
> | Evo-Step-Mistral-7B    | 72.65% |82.06%      | 52.61%         | **40.26%**      |
> | LLaMA-3-8B     | 0%         | 0%              | 0%                  | 0%              |
> | Evo-Step-LLaMA-3-8B      | **84.49%** | **85.28%**      | **61.61%**         | 36.36%     |
>
> We hope this explanation addresses the reviewer’s concerns.
>
> >Question 4. What would be the impact of directly applying the proposed "Stepwise Validation Mechanism" to SOTA LLMs, such as GPT-4, in modeling tasks? Would this significantly enhance accuracy and performance?
>
> To evaluate the impact of the proposed Stepwise Validation Mechanism on SOTA LLMs, including GPT-3.5, GPT-4 and GPT-4o, we conducted experiments on the MAMO Complex dataset. The results are summarized below:
>
> **Table 9: Comparison of Stepwise Validation Mechanism and Other Prompt Engineering Methods on MAMO ComplexLP**
>
> | Method\Model |GPT-3.5|GPT-4|GPT-4o|
> |----------------------------|-------------|-----------|------------|
> | Standard | 10.90%| 24.64%| 46.92%|
> | CoT| 13.27%| 29.86%| 49.29%|
> | Reflexion| 14.22%| 36.02%| 48.34%|
> | CoE| **17.06%**| 40.28%| **54.03%** |
> | Stepwise Validation Mechanism| 16.59%|**42.18%**| 50.71%|
>
> The results demonstrate that the Stepwise Validation Mechanism delivers consistent improvements across different GPT models when compared to Standard, CoT, and Reflexion methods. For GPT-4, our framework achieves the highest accuracy (42.18%), outperforming all other methods. However, CoE remains superior for GPT-3.5 and GPT-4o, reflecting the strength of its iterative reflection mechanism in these cases.
>
> In contrast, the Stepwise Validation Mechanism emphasizes real-time validation and correction during the modeling process, , avoiding the additional complexity of reflection. This streamlined approach proves particularly effective for LLMs, as demonstrated by its superior performance with GPT-4. Although CoE excels in certain cases, our method offers a robust and efficient alternative.
>
>  Additionally, It is important to consider the inherent difficulty of solving tasks directly during testing, as all methods must generate solutions from scratch. However, when used for data generation, the Stepwise Validation Mechanism can reference the solution of the original problem to generate solutions for new problems. By focusing only on the newly added or modified components, the mechanism significantly reduces the modeling difficulty. This advantage is not available during testing, where tasks must be solved entirely independently, but it underscores the potential of Stepwise Validation Mechanism for facilitating high-quality data generation.

---

> > ### Comment · Reviewer_Je8b · 2024-11-22
> >
> > I appreciate the responses and additional experimental results from the authors. I think the additional experiments have validated the effectiveness, and my main concerns have been addressed. I will raise my score accordingly.

---

### Author Response · Authors · 2024-11-25
**General Response to Reviewers and Submitted Revisions**

We would like to express our sincere gratitude to all the reviewers for their valuable feedback and constructive suggestions.

The manuscript has been revised, with changes highlighted in blue text in the PDF. Below is a summary of the key revisions:

1) The abstract has been rewritten to emphasize that the focus of the work is on modeling operational research problems.

2) The term "Evolutionary generation" has been replaced with "iterative problem generation" to avoid confusion.

3) The more general term "evolving" has been used to replace specialized terminology.

4) Detailed captions have been added to the images in the article to enhance readability.

Additionally, modifications have been made to the core sections of the article to improve clarity and precision. Furthermore, we have made the generated dataset, code, and testing results publicly available to facilitate reproducibility.


We sincerely appreciate the reviewers' thoughtful comments and would like to thank you again for your valuable feedback. We hope the revised manuscript and responses address your concerns. Please feel free to reach out if you have any further questions or require any additional details. We are more than happy to provide further clarification or additional information if needed.

---

### Meta-Review · Area_Chair_qxLw · 2024-12-20

**Metareview:**

Summary:
This paper proposes an evolutionary framework for data generation aimed at training LLMs for operations research optimization modeling. The data generation process involves two key components: an evolutionary approach for instance generation and a subsequent step for solution assessment and refinement. The framework is evaluated on three OR datasets with varying levels of difficulty, demonstrating promising performance.

Strengths:
The proposed framework automates the data generation process, improving efficiency and effectiveness. Moreover, the paper is well-structured and easy to follow.

Weakness:
The paper does not include comparisons with non-LLM heuristics, as highlighted by one of the reviewers.

Decision:
While this paper presents an interesting idea, the lack of comparison with non-LLM heuristics undermines the evaluation of its true effectiveness. As a result, it falls into a borderline category. I recommend rejection and suggest that the authors address this issue to improve their chances for future submissions.

**Additional Comments On Reviewer Discussion:**

Two out of the four reviewers voted for rejection after rebuttal. While the authors addressed many concerns raised by the reviewers, one of the reviewers insisted the necessary comparison with the non-LLM heuristics. While I believe it is a borderline submission, I also agree that the comprehensive experimental comparison is important to meet the ICLR standard, and I recommend rejection.

---

### Decision · Program_Chairs · 2025-01-22

Reject